# Differences in subcortico-cortical interactions identified from connectome and microcircuit models in autism

Bo-yong Park[1,2✉], Seok-Jun Hong [1,3,4,5], Sofie L. Valk[6,7], Casey Paquola [1], Oualid Benkarim [1], Richard A. I. Bethlehem [8,9], Adriana Di Martino[3], Michael P. Milham [3], Alessandro Gozzi[10], B. T. Thomas Yeo [11,12,13,14,15], Jonathan Smallwood [16,17] & Boris C. Bernhardt [1✉]

The pathophysiology of autism has been suggested to involve a combination of both macroscale connectome miswiring and microcircuit anomalies. Here, we combine connectome-wide manifold learning with biophysical simulation models to understand associations between global network perturbations and microcircuit dysfunctions in autism. We studied neuroimaging and phenotypic data in 47 individuals with autism and 37 typically developing controls obtained from the Autism Brain Imaging Data Exchange initiative. Our analysis establishes significant differences in structural connectome organization in individuals with autism relative to controls, with strong between-group effects in low-level somatosensory regions and moderate effects in high-level association cortices. Computational models reveal that the degree of macroscale anomalies is related to atypical increases of recurrent excitation/inhibition, as well as subcortical inputs into cortical microcircuits, especially in sensory and motor areas. Transcriptomic association analysis based on postmortem datasets identifies genes expressed in cortical and thalamic areas from childhood to young adulthood. Finally, supervised machine learning finds that the macroscale perturbations are associated with symptom severity scores on the Autism Diagnostic Observation Schedule. Together, our analyses suggest that atypical subcortico-cortical interactions are associated with both microcircuit and macroscale connectome differences in autism.

[1] McConnell Brain Imaging Centre, Montreal Neurological Institute and Hospital, McGill University, Montreal, QC, Canada. [2] Department of Data Science, Inha University, Incheon, South Korea. [3] Center for the Developing Brain, Child Mind Institute, New York City, NY, USA. [4] Center for Neuroscience Imaging Research, Institute for Basic Science, Sungkyunkwan University, Suwon, South Korea. [5] Department of Biomedical Engineering, Sungkyunkwan University, Suwon, South Korea. [6] Forschungszentrum, Julich, Germany. [7] Max Planck Institute for Cognitive and Brain Sciences, Leipzig, Germany. [8] Autism Research Centre, Department of Psychiatry, University of Cambridge, Cambridge, UK. [9] Brain Mapping Unit, Department of Psychiatry, University of Cambridge, Cambridge, UK. [10] Istituto Italiano di Tecnologia, Centre for Neuroscience and Cognitive Systems @ UNITN, Rovereto, Italy. [11] Department of Electrical and Computer Engineering, National University of Singapore, Singapore, Singapore. [12] Centre for Sleep and Cognition (CSC) & Centre for Translational Magnetic Resonance Research (TMR), National University of Singapore, Singapore, Singapore. [13] N.1 Institute for Health & Institute for Digital Medicine (WisDM), National University of Singapore, Singapore, Singapore. [14] Martinos Center for Biomedical Imaging, Massachusetts General Hospital, Charlestown, MA, USA. [15] Integrative Sciences and Engineering Programme (ISEP), National University of Singapore, Singapore, Singapore. [16] Department of Psychology, York Neuroimaging Centre, University of York, York, UK. [17] Department of Psychology, Queen's University, Kingston, ON, Canada. ✉email: bo.y.park@mcgill.ca; boris.bernhardt@mcgill.ca

Autism is one of the most common neurodevelopmental conditions, with persistent impairments that challenge affected individuals and their families, as well as health care and educational systems at large[1–3]. Despite extensive research efforts, the conceptualization and management of autism continue to face significant challenges. A major difficulty in the neurobiological understanding of autism is that the condition appears to impact multiple scales of brain organization[4–10]. Contemporary studies suggest that autism is characterized by atypical connectivity at macroscale[6–8,10–14], alongside with local changes in cortical microcircuit function such as excitation/inhibition imbalance[4,5,9,15–17]. However, an overarching framework of how these microscale findings relate to autism-related brain reorganization at macroscale remains to be established.

Neuroscience has recently gained unprecedented opportunities to interrogate the living human brain at multiple scales in both health and disease[4,6,8,9,18], particularly through advances in multimodal neuroimaging. A wealth of studies have examined changes in cortical morphology[12,19], as well as atypical functional connectivity[6,10,14,20–25], in individuals with autism relative to typically developing controls. On the other hand, less is known about macroscopic changes in structural connectivity[26–29]. Captializing on diffusion magnetic resonance imaging (dMRI) and tractographic reconstructions of structural wiring[30–35], previous studies observed alterations in diffusivity parameters and connectivity strength of several inter-regional fiber pathways in autism[27,28]. Beyond the analysis of specific fiber tracts, macroscale brain organization is increasingly studied using connectome-wide analyses. One promising approach is the application of manifold learning techniques that project high dimensional connectomes into low dimensional representations. These methods can represent continuous changes in connectivity, and can incorporate multiple, potentially overlapping, connectivity gradients along the cortical surface[36,37]. As such, they complement widely used parcellation approaches that place discrete boundaries between regions and that average connectivity measures within each parcel, which may potentially mix the signals from different large-scale gradients[37]. In neurotypical individuals, these techniques have gained traction to study brain connectivity and cortical microstructure, and to represent complex neural organizaton in a compact analytical space[6,38–41]. This approach, however, remains underexplored in the assessment of atypical structural wiring in autism.

In addition to providing a synoptic perspective on macroscale structural wiring, features of structural connectivity can also be used to predict functional dynamics to understand how the structural organization of the brain determines its function[18,42–45]. One class of methods simulates whole-brain functional dynamics via a network of anatomically connected neural masses[43–45]. In contrast to approaches that assess structure–function coupling statistically[46–49], these models are governed by biophysically plausible parameters that are anchored in established models of neural circuit function[18,42]. A recent study in healthy young adults established that these models robustly simulate intrinsic functional networks from structural connectivity data, and the study incorporated model inversion approaches to estimate regionally varying microcircuit parameters, specifically recurrent excitation/inhibition and external subcortical input into cortical microcircuits[18]. Applying these models to autism, therefore, may provide the opportunity to understand how microcircuit-level features correlate with macroscale connectivity patterns.

Here, we show macro- and microscale perturbations in individuals with autism relative to typically developing controls, and examine their relationship. We generate low dimensional representations of structural connectomes by applying manifold learning techniques to dMRI data[41,50], and use these to build a macroscale account of topographical structural divergence in autism. We also apply biophysical computational simulations to infer microcircuit-level imbalances at a regional level, specifically recurrent excitation/inhibition and excitatory subcortical input[18]. We embed our results in a neurobiological and neurodevelopmental context by spatially correlating the macroscale patterns with postmortem maps of gene expression data[51–53]. Finally, we establish associations between our macroscale findings and autism symptom severity using supervised machine learning with five-fold nested cross validation[54–57].

## Results

Our sample consisted of 47 individuals with autism and 37 neurotypical controls obtained from the two independent sites (Supplementary Table 1 for demographic information) from the Autism Brain Imaging Data Exchange initiative (ABIDE-II; https://fcon_1000.projects.nitrc.org/indi/abide)[58,59]. See Methods for details on participant selection, image processing, manifold generation, computational modeling, transcriptomic analysis, and symptom prediction.

**Large-scale structural connectome manifolds**. We estimated a cortex-wide structural connectome manifold using nonlinear diffusion map embedding (https://github.com/MICA-MNI/BrainSpace)[41]. The template manifold was estimated from an unbiased and group representative structural connectome[60] to which individual manifolds were aligned (see Methods)[41,61]. The three dimensions (henceforth, M1, M2, and M3) reflect the principle axes of variation in structural connectivity accounting for approximately 50.6% of the total variance (Fig. 1a). Each cortical region can be described in terms of its position along these three axes. The individual dimensions extended from somato-motor to visual areas (M1), differentiated lateral parietal/motor cortex from prefrontal cortex (M2), and showed a lateral to medial cortical axis (M3; Fig. 1b).

**Connectome manifold distortions in autism**. Cortex-wide multivariate analyses compared connectome manifolds spanned by M1–M3 between individuals with autism and controls, using a model that additionally controlled for age, sex, and site. Relative to controls, we observed macroscale distortions in autism in multiple networks, with primary effects in sensory and somatomotor as well as heteromodal association cortices (false discovery rate (FDR) < 0.05; Fig. 1c). Stratifying effects according to a seminal model of neural organization that contains four cortical hierarchy levels (1: idiotypic; 2: unimodal association; 3: heteromodal association; 4: paralimbic; Supplementary Table 2)[62], we identified peak effects in idiotypic areas followed by unimodal and heteromodal association cortices. Similarly, when analyzing effects with respect to seven intrinsic functional communities[63], the strongest between-group differences were observed in somatomotor networks followed by higher-order systems, such as the default-mode network (Supplementary Fig. 1a). To address the lateralization of findings, we stratified the between-group effects according to cortical hierarchical levels[62] for left and right hemispheres separately (Supplementary Fig. 1b). We found 49% stronger effects in the right compared to the left idiotypic networks and 29% stronger effects in left versus right for heteromodal association cortices. A comparable effect was seen when stratifying findings across intrinsic functional communities[63].

When we summarized the multivariate manifolds into a single scalar that represents manifold expansion and contraction[64,65], we found evident contractions in somatomotor and posterior cingulate cortices, and expansions in heteromodal association

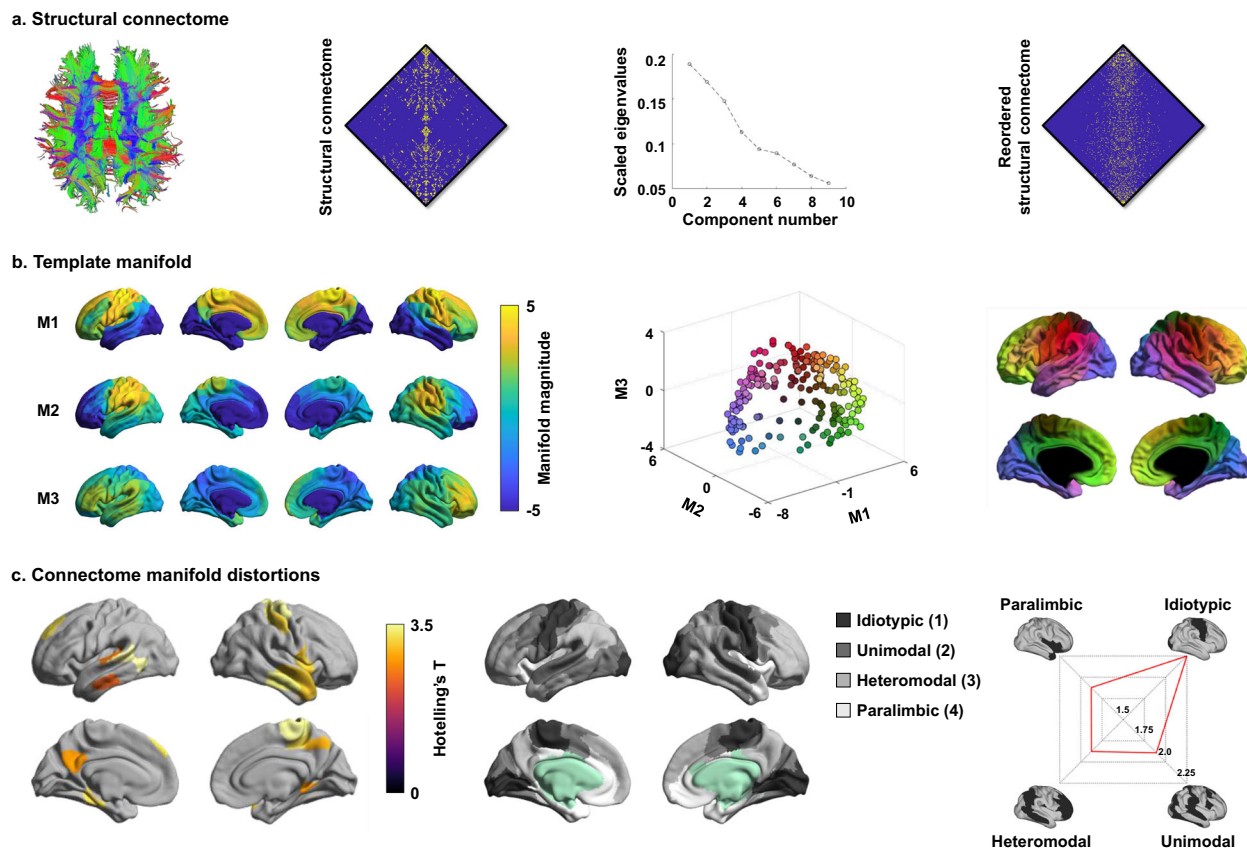

**Fig. 1 Structural connectome manifolds. a** Fiber tracts generated from dMRI (left), a cortex-wide structural connectome (middle left), and a scree plot describing connectome variance after identifying principal eigenvectors (middle right). The structural connectome reordered according to M1 is shown for better visualization (right). **b** Manifolds estimated from the structural connectome (left). Three dimensions (M1, M2, and M3) explained >50% of variance and corresponded to the clearest eigengap. Each data point (i.e., brain region) was represented in the three-dimensional manifold space with different colors (middle), and was mapped onto the brain surface for visualization (right). **c** The t-statistics of the identified regions that showed significant between-group differences in these dimensions between individuals with autism and controls (left). Findings have been corrected for multiple comparisons at false discovery rate (FDR) < 0.05. Stratification of between-group difference effects along cortical hierarchical levels (middle)[62] is presented in the radar plot (right). Source data are provided as a Source Data file.

cortex in individuals with autism relative to controls (Supplementary Fig. 1c), and these patterns were similar across both sites (Supplementary Fig. 1d). To ensure that our results were not related to spurious features, we assessed the degree of head motion of each individual during the dMRI scan based on framewise displacement (FD), and found that mean FD did not differ between autism and controls ($p = 0.34$) (Supplementary Fig. 2a). Notably, between-group differences in structural manifolds were comparable when controlling for mean FD, indicating that head motion did not considerably affect patterns of structural connectome perturbations in autism (Supplementary Fig. 2b). Repeating analyses separately in children (age < 18) and adults (age ≥ 18), effects in adults with autism were highest in higher-order frontoparietal/paralimbic areas while children with autism displayed most substantial anomalies in somatomotor/idiotypic regions (Supplementary Fig. 3), similar to age-stratified results in a previous functional connectome study[6].

Prior MRI research has indicated atypical cortical morphology in autism, showing anomalies in both cortical thickness and folding relative to controls[12,19], motivating an assessment of morphological effects on manifold findings. Correlating manifold distortions with cortical thickness and folding variations, we observed only marginal relations ($p = 0.1$; Supplementary Fig. 4a). In addition, connectome manifold differences between autism and controls were still measurable when controlling for cortical thickness and curvature in the same model, indicating that

structural connectome perturbations occurred above and beyond any potential variations in cortical morphology (Supplementary Fig. 4b).

We also compared each manifold dimension between individuals with autism and controls. We found that the first dimension showed significant effects in lateral temporal regions and the second and third dimensions showed significant effects in medial temporal and lateral somatosensory areas (Supplementary Fig. 5). Patterns were similar to the multivariate findings, but effects were stronger when considering all dimensions simultaneously.

To explore specific connections that potentially contribute to the above manifold distortions, we compared the streamline cross-section between individuals with autism and controls. We could observe a mix of connectivity increases and decreases in autism, affecting multiple networks, including those in which we observed manifold anomalies (Supplementary Fig. 6a, b). Notably, the lengths of fiber tracts showing decreases were considerably longer (20.90 mm) compared to fiber tracts showing increases (9.37 mm) ($p < 0.001$; Supplementary Fig. 6c). To further assess subcortico-cortical and cerebello-cortical connectivity, we projected the streamline strength of these regions to cortical targets in each individual to the manifold space (see Methods). For each subcortical and cerebellar structure, multivariate analysis compared weighted manifolds (wM) spanned by wM1–wM3 between individuals with autism and controls while controlling for age, sex, and site. After FDR-correction, we found significant between-

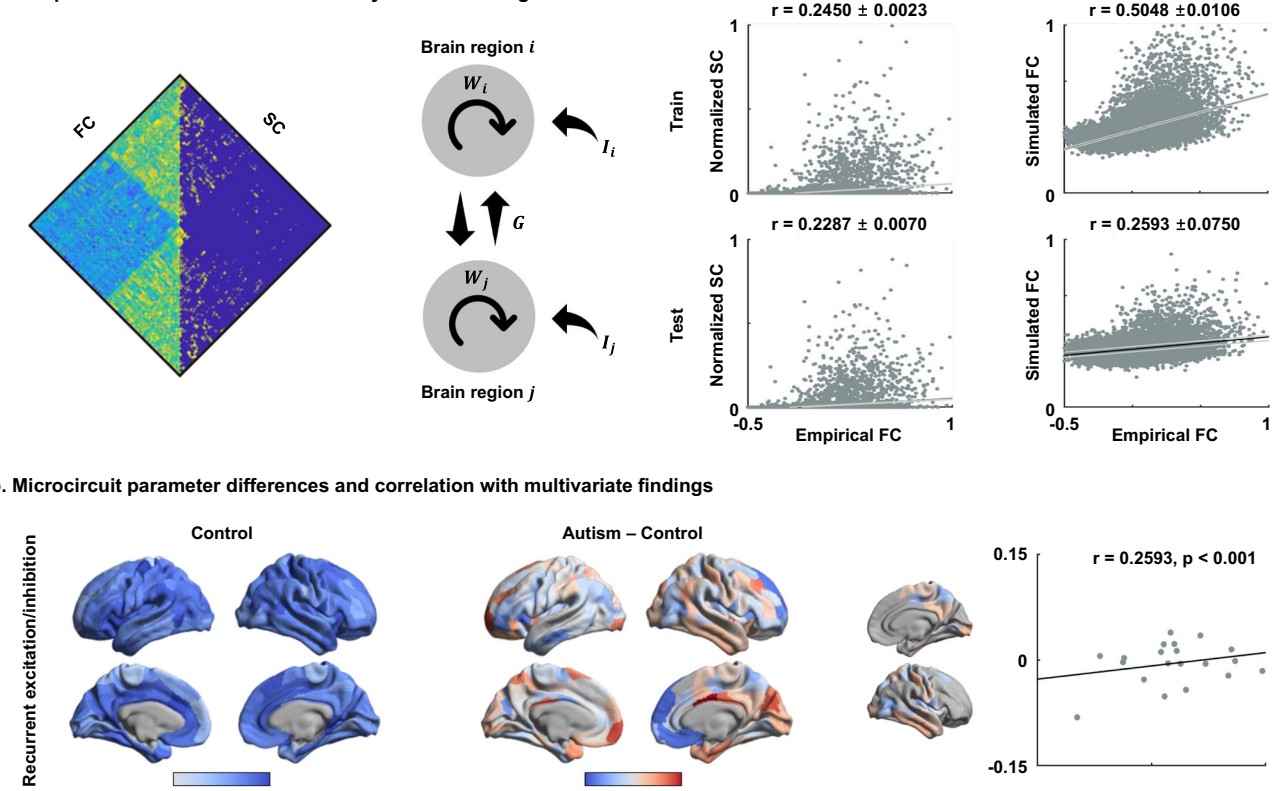

**Fig. 2 Microcircuit parameters and associations with macroscale findings. a** A relaxed mean-field model[18] was used to predict functional connectivity (FC) from structural connectivity (SC) and to estimate region-specific microcircuit parameters, i.e., recurrent excitation/inhibition $W_i$ and subcortical/external input $I_i$ (left). A global coupling constant $G$ is also estimated. Linear correlations between FC and SC, and empirical and simulated FC are shown (right). Black lines indicate mean correlation and gray lines represent 95% confidence interval across cross validation. **b** Microcircuit parameters of controls (left) and differences of the parameters between individuals with autism and controls (middle). Linear correlations between *t*-statistics derived from the multivariate group comparison and the regional changes in microcircuit parameters are reported, constrained to regions showing significant between-group differences in Fig. 1c (right). Source data are provided as a Source Data file.

group differences for the thalamus (FDR = 0.03), caudate (FDR = 0.04), and cerebellum (FDR = 0.03), indicating that the low dimensional representation of subcortico-cortical connectivity patters of these regions changed in autism (Supplementary Fig. 7).

**Microcircuit parameters from biophysical network modeling.** Biophysical computational simulations[18] complemented our macroscale findings by modeling atypical microcircuit-level functional dynamics. Harnessing a relaxed mean-field model[18], we simulated dynamics of functional signals through a set of simplified nonlinear stochastic differential equations by linking ensembles of local neural masses (i.e., theoretical cell population models for excitatory neurons, which reciprocally inhibit each other; Fig. 2a) with diffusion-

derived structural connectivity. Notably, the model iteratively tunes its parameters to simulate functional connectivity patterns that are maximally similar compared to empirical data, which also resulted in an optimal set of biophysical parameters (i.e., recurrent excitation/inhibition and excitatory subcortical/external input; see Methods). We ran the mean-field model[18] with five-fold cross validation to first evaluate the capacity of the structural connectome to simulate intrinsic functional dynamics, and then estimated regional microcircuit parameters. The optimal model predicted functional connectivity (product-moment correlation coefficient r ~0.5 for training and r ~0.26 for test data) nominally higher than the corresponding baseline correlations between structural and functional connectivity (r ~0.25 for training and r ~0.23 for test data) (Figs. 2a and S8a). We assessed improvements in predicting

functional connectivity with the biophysical model compared to a baseline model based on structural connectivity with three different approaches. First, we observed that the model-driven correlations (i.e., between empirical and simulated functional connectivity) were higher than baseline correlations (i.e., between structural and empirical functional connectivity) in all 5/5 folds for both training and test data. Second, we performed 1000 permutation tests, in which we randomly assigned elements of structural and simulated functional connectivity (see Methods). We found that the empirical and simulated functional connectivity showed significantly higher correlation compared to structural and empirical functional connectivity ($p < 0.001$ for both training and test data). Finally, we ran 1000 bootstraps using dimensionality reduced structural and functional connectomes based on functional communities[63] (see Methods). The correlation coefficients between empirical and simulated functional connectivity exceeded corresponding baseline correlations between structural and functional connectivity (Supplementary Fig. 8b; $p < 0.001$ for both training and test). Together, these findings suggest that the biophysical model provided improvements in predicting functional connectivity relative to using baseline structural connectivity. Estimated microcircuit parameters (Fig. 2b) were relatively stable across cross-validations in terms of the variance, with a mean ± SD of the parameter values across the five folds of $0.530 ± 0.004$ for recurrent excitation/inhibition and $0.325 ± 0.001$ for subcortical/external input in controls, and $0.528 ± 0.006$ for recurrent excitation/inhibition and $0.325 ± 0.001$ for subcortical/external input in autism. The product-moment correlations of the microcircuit parameters across the cross-validation folds were mean ± SD of $0.79 ± 0.04$ for recurrent excitation/inhibition and $0.91 ± 0.03$ for subcortical/external input for controls, and $0.74 ± 0.05$ and $0.89 ± 0.03$ for individuals with autism, indicating robustness. To confirm stability, we ran a bootstrap-based evaluation of the relaxed mean-field model based on an intrinsic functional community partitioning[63] (see Methods). The mean ± SD of the parameters across 1000 bootstraps in autism relative to controls were consistent (Supplementary Fig. 8c). Between-group differences in microcircuit parameters between autism and controls were assessed using 1000 permutation tests. We found increased recurrent excitation/inhibition in visual ($p = 0.02$) and limbic networks ($p < 0.001$) in autism; considering subcortical/external inputs, we observed decreases in dorsal attention ($p < 0.001$), frontoparietal ($p = 0.02$), and default-mode networks ($p = 0.01$), while values in sensorimotor networks increased ($p < 0.001$) (Supplementary Fig. 8c). Estimated microcircuit parameters were largely consistent across different matrix thresholds (Supplementary Fig. 9). Notably, between-group differences in parameters did not show significant differences across different thresholding procedures, suggesting sensitivity. Correlating these regional changes in microcircuit patterns (see Fig. 2b) with multivariate macroscale manifold anomalies (see Fig. 1c), we observed a significant correlation between the overall degree of manifold distortion and increases in excitation/inhibition ($r = 0.26$, $p < 0.001$; determined using nonparametric spin tests that account for spatial autocorrelation[41,66]), as well as increases in excitatory subcortical/external input ($r = 0.20$, $p = 0.02$) (Fig. 2b).

**Transcriptomic association analysis**. We next performed transcriptomic association analysis and developmental and disease enrichment analyses to explore neurobiological underpinnings of the macroscale manifold findings in autism identified in our analysis (Fig. 3a). Specifically, we correlated the multivariate change pattern with postmortem gene expression data of six donors from the Allen Institute for Brain Sciences (AIBS)[67,68]. For significant gene lists after multiple comparisons correction (FDR <

0.05), we repeated the transcriptomic association analysis with randomly rotated maps of the multivariate change pattern for 100 times, to ensure that genes were not selected by chance. Among the significantly associated genes, we selected only those that were consistently expressed across donors ($r > 0.5$)[69] (Supplementary Data 1). We fed those into a developmental gene expression analysis, which highlights developmental time windows across brain regions in which these genes are expressed (see Methods)[53]. This analysis highlighted associations between the multivariate pattern of autism-related structural manifold distortions and genes expressed in early childhood and adolescence, as well as early infancy and young adulthood, in thalamic and cortical areas (Fig. 3b). While these genes were also expressed in the cerebellum in early development and in the amygdala in later developmental stages, they were not significantly expressed in the striatum nor hippocampus. We furthermore validated the transcriptomic association results using the Genotype-Tissue Expression (GTEx) database (https://www.gtexportal.org/home), and we found that the genes highly associated with multivariate manifold changes were strongly expressed in cortical areas, replicating our results (Supplementary Fig. 10a). In addition, we performed disease enrichment analysis to associate the significance of the gene expressions with the log fold-changes of autism, schizophrenia, and bipolar disorder (see Methods)[70]. Notably, autism showed the most marked associations (T = $-34.89$ and $p < 0.001$) followed by schizophrenia (T = $-8.93$ and $p < 0.001$) and bipolar disorder (T = $5.34$ and $p < 0.001$) (Fig. 3c).

We further compared the genes associated with multivariate connectome manifold changes with distinct cell types proposed in prior work[71,72]. For each cell type, we calculated the overlap ratio, which indicates how many genes expressed for manifold changes are included in each cell-type-specific genes (Supplementary Fig. 10b). Cell-type-specific expression analysis indicates several cell types showing a similar expression profile with positive overlap ratio. Highest and marginally significant overlap was observed for the excitatory neurons (mean ± SD across 13 cell subtypes = $22.36 ± 4.16\%$; FDR = 0.1; see Methods) relative to others (endothelial cells: 14.29%; astrocytes: 11.11%; inhibitory neurons: mean ± SD across 11 cell subtypes = $7.47 ± 4.21\%$; pericytes, oligodendrocytes and their precursor cells, microglia: 0%).

**Associations to symptom severity**. We used supervised machine learning to predict symptom severity scores based on the Autism Diagnostic Observation Schedule (ADOS—social cognition, communication, and repeated behavior/interest subscores and total score)[73] using structural connectome manifold information. Specifically, we employed elastic net regularization[74] with five-fold nested cross validation (see Methods)[54–57]. The procedure was repeated 100 times with different training and test data compositions to avoid subject selection bias. Using a regularization parameter of 0.6, manifolds spanned by M1–M3 significantly predicted total ADOS score (mean ± SD $r = 0.47 ± 0.06$; mean ± SD mean absolute error (MAE) = $2.07 ± 0.12$; permutation test $p = 0.01$) as well as subscores for social cognition (mean ± SD $r = 0.43 ± 0.07$; MAE = $1.79 ± 0.09$; $p = 0.01$), communication (mean ± SD $r = 0.57 ± 0.03$; MAE = $0.89 ± 0.03$; $p < 0.001$), and marginally for repeated behavior/interest (mean ± SD $r = 0.33 ± 0.09$; MAE = $0.89 ± 0.05$; $p = 0.09$) (Fig. 4). Features were selected in premotor, lateral prefrontal and orbitofrontal, lateral and medial temporal, lateral parietal, and posterior cingulate regions. (for results based on other regularization parameters, see Supplementary Table 3).

Performing the same analyses using three-fold nested cross validation, we found largely prediction results (Supplementary Table 4). Indeed, the regularization parameter of 0.7 and 0.8 showed good performance for predicting ADOS scores, and features were

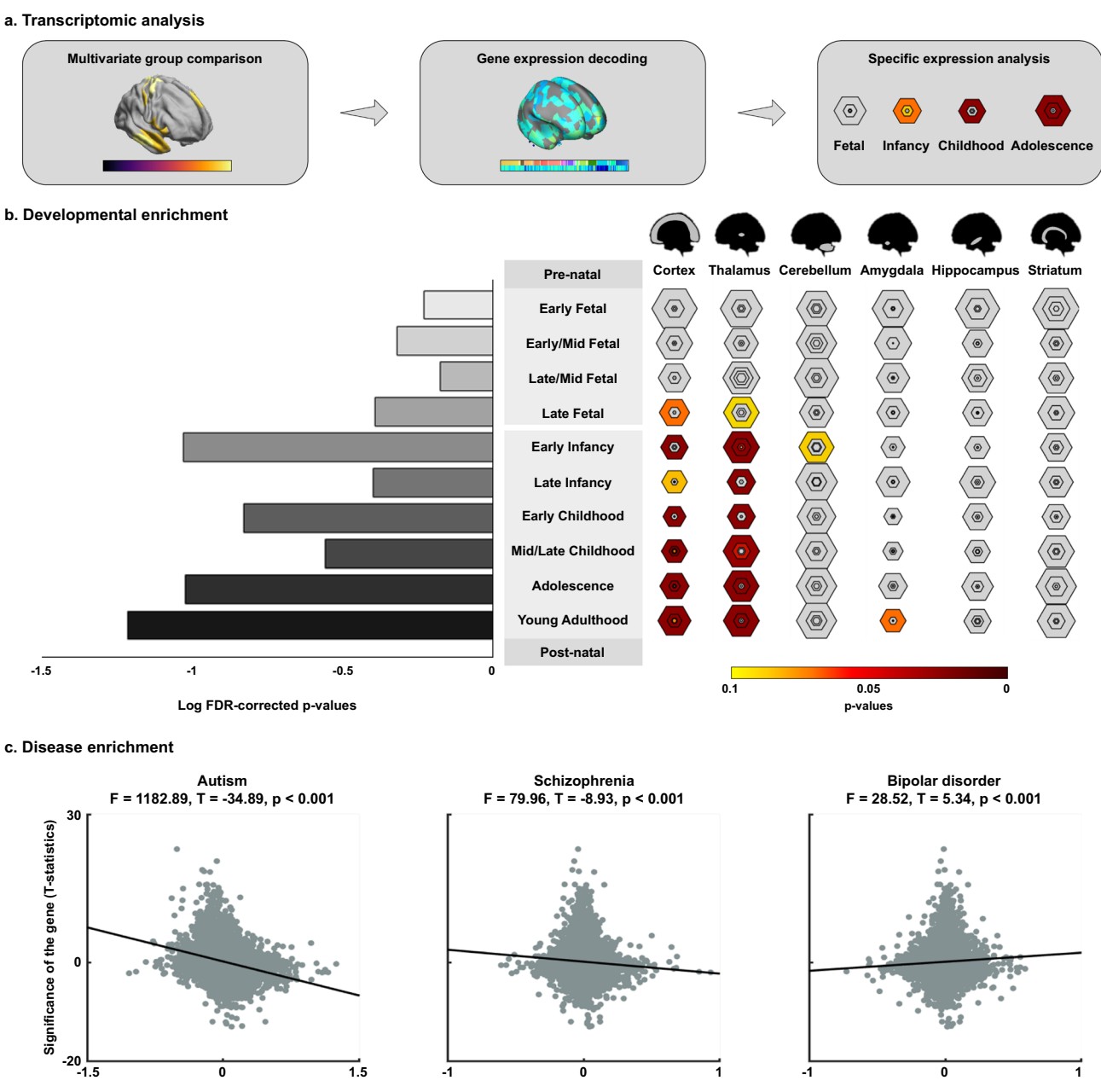

**Fig. 3 Transcriptomic analysis to identify gene expression patterns. a** Schema to relate multivariate manifold distortions with gene expression patterns and to perform a cell-type-specific gene expression analysis. **b** Developmental enrichment, showing strong associations with cortex and thalamus during early childhood and adolescence, as well as early infancy and young adulthood. The size of hexagon rings represents the proportion of genes specifically expressed in a particular tissue at a particular developmental stage. Varying stringencies for enrichment with respect to specificity index threshold (pSI) are represented by the size of hexagons going from least specific (outer hexagons) to most specific (center hexagons) (pSI = 0.05, 0.01, 0.001, and 0.0001, respectively)[53]. Colors represent the false discovery rate (FDR)-corrected p-values. The bar plot on the left represents the log-transformed FDR-corrected *p*-values, averaged across all brain structures. **c** Disease enrichment analysis for associating gene expressions with disease effects of autism, schizophrenia, and bipolar disorder. Source data are provided as a Source Data file.

primarily selected in lateral and medial prefrontal, lateral parietal, and lateral temporal regions. We repeated symptom severity prediction for each site separately (Supplementary Table 5 and 6). Although each site contains small number of subjects ($n = 20$ for New York University Langone Medical Center (NYU), $n = 18$ for Trinity College Dublin (TCD)), we found consistent prediction results with similar optimal regularization parameters. Lastly, we performed symptom severity prediction using the edge values of the structural connectivity matrix (i.e., streamline cross-section; Supplementary Table 7). Similar to the results from connectome

manifolds, regularization parameter of 0.6 and 0.7 showed good performance for predicting ADOS scores, where the selected connections were primarily found in medial prefrontal, lateral parietal, lateral temporal, posterior cingulate, and sensorimotor regions.

## Discussion

Understanding autism pathophysiology remains challenging, in part, because of the difficulties in consolidating neuroimaging

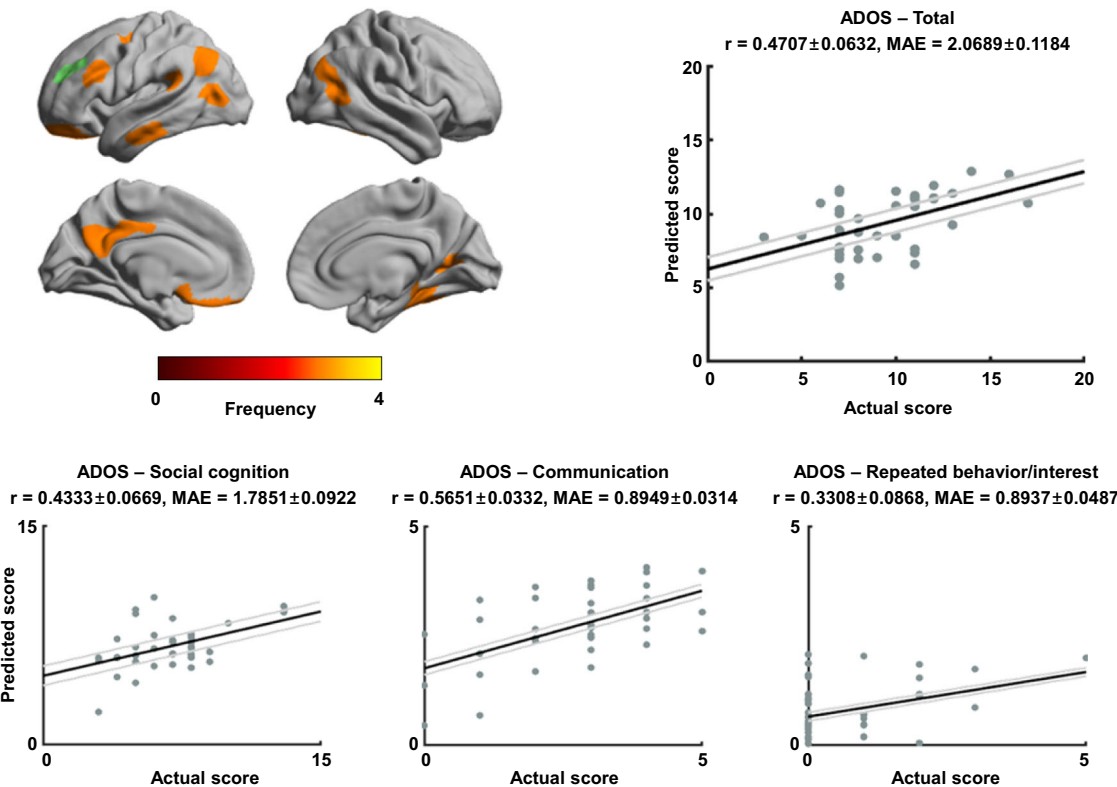

**Fig. 4 Associations between structural manifolds and autism symptoms.** Frequency of the selected brain regions across four ADOS scores reported on brain surfaces (top left). Correlations between actual and predicted ADOS total and subscores are reported with scatter plots. Black line indicates mean correlation and gray lines represent 95% confidence interval for 100 iterations with different training/test dataset. Source data are provided as a Source Data file. ADOS Autism Diagnostic Observation Schedule, MAE mean absolute error.

findings of connectome miswiring with molecular and neurophysiological data that probe cortical microcircuits. By combining manifold learning and computational models of brain dynamics, our study established how macroscale structural connectome alterations in autism relate to microcircuit dysfunction. We identified macroscale changes in cortical networks in autism, with peak differences in somatosensory as well as heteromodal association cortex, particularly within the posterior core of the default-mode network. Findings were broadly similar when controlling for head motion and morphological changes, such as cortical thickness and curvature, and were consistent across different study sites. Using biophysical parameters derived from a large-scale computational model, we found that these whole-brain findings were correlated to alterations in subcortical drive into cortical microcircuits, together with alterations in excitation/inhibition. An association with subcortical structures, particularly the thalamus, was supported by complementary analyses of subcortical connectivity and postmortem transcriptomic association and developmental as well as disease enrichment analyses. These analyses highlighted that the affected regions harbor genes expressed in cortical and thalamic areas in early childhood and adolescence, as well as early infancy and young adulthood. Our findings, therefore, offer a perspective on the relation between subcortico-cortical interactions at macroscale and microcircuit reorganization in autism.

The current work utilized manifold learning to compress high dimensional structural connectomes into a series of principal axes that describe spatial trends in connectivity changes across the cortical mantle in a data-driven manner. By offering a cortex-wide analysis of structural connectivity, our work extends prior diffusion MRI studies in autism that have shown atypical microstructure in

fiber tracts interconnecting higher-order brain systems[13,29,75–77] and work focusing on fibers mediating connectivity between sensorimotor and subcortical systems[7,78]. Our findings are also consistent with prior graph-theoretical studies of structural connectome data in autism that highlight alterations in global as well as local efficiency across both lower-level and higher-order cortical systems[29,78–82]. In parallel, our findings provide insights on potential structural substrates underlying a wide range of functional network anomalies reported in autism[6,83,84]. Functional findings are somewhat heterogeneous across studies and analytical approaches; yet prior studies have converged on an overall pattern characterized by cortico-cortical functional connectivity reductions, often affecting heteromodal association cortices, such as the default-mode network, together with patches of connectivity increases, particularly between sensorimotor cortices and subcortical nodes such as the thalamus[7,11,85,86]. By highlighting both association cortices, such as the default-mode network, as well as idiotypic and somatosensory systems, our work provides a potential consolidation of these distributed effects in a space governed by structural wiring. Notably, connectome manifold distortions in autism showed more marked effects in the right hemisphere for idiotypic areas, and more marked effects in the left hemisphere in heteromodal areas. Several prior studies suggested asymmetric findings in autism at the level of cortical morphology as well as functional organization[87–93]. Our results may contribute to these prior findings by suggesting a differential susceptibility for structural reorganization in autism for left and right hemispheres for unimodal versus heteromodal regions, respectively.

Histological studies have suggested several potential cellular substrates associated with connectome miswiring of autism, including altered cortical lamination[94–97] and columnar

layout[98,99], together with atypical neuronal migration that can result in cortical blurring[95,100] and changes in spine density of cortical projection neurons[101,102]. Such cellular changes may impact the functional organization of cortical microcircuits in autism, a possibility supported by molecular studies in animals[4,22–25,103,104]. These findings collectively give rise to the notion that cortical areas may show imbalances in excitation and inhibition in autism[4,9,16,22,105,106]. Such imbalances have been related to anomalies in cortical neurotransmitter systems[107–110] and atypical subcortico-cortical interactions, where subcortical structures such as thalamus are thought to serve as modulators[7,11,24,85,86,111]. Our study provides further support for the role of perturbations in cortical microcircuit function in autism from a network perspective, by leveraging a biophysically plausible computational model of brain function, which tunes parameters to optimize the link between structural and functional connectomes[18]. In recent work[18], these models were inverted to infer regional variations in cortical microcircuit parameters. Studying a cohort of healthy adults from the human connectome project dataset, the study mapped cortex-wide gradients of recurrent excitation/inhibition and of excitatory subcortical input, with a topography mirroring prior work showing gradients in laminar differentiation and synaptic organization in nonhuman primates[62,112]. In the neurotypical individuals studied here, we found spatial trends in excitation/inhibition and subcortical input that resembled those reported in the recent study, showing increased subcortical input but lower excitation/inhibition in association cortices, while sensorimotor areas showed higher excitation/inhibition. This correspondence is an important consideration given that the retrospective data aggregated and shared via ABIDE is not at par in terms of image quality and data volume with the human connectome project data on which these models were originally presented[18,113]. Importantly, comparing microcircuit maps between individuals with autism and neurotypical controls suggested a relatively diffuse pattern of local microcircuit parameter changes, characterized by both reductions as well as increases. In healthy individuals, inter-regional variations in recurrent excitation/ inhibition and subcortical/external input follow sensory-fugal hierarchical gradients, previously established with resting-state functional connectivity mapping and analysis of myelin-sensitive MRI contrasts[18]. Our computational modeling findings, showing atypical microcircuit parameters in somatosensory but also higher-order default-mode networks in individuals with autism relative to controls, suggest a broad microcircuit imbalance in autism that is not limited to a specific network. Our results also indicate that these microcircuit imbalances impact multiple stages of the cortical hierarchy, a pattern consistent with prior resting-state fMRI analysis, and with the broad phenotypical correlates of autism that encompass both sensory deficits as well as atypical features of higher-order cognition[6,8,16,114]. In our work, directly correlating connectome-wide manifold distortions with microcircuit parameters indicated associations with increased recurrent excitation/inhibition and excitatory subcortical drive, a finding especially marked in idiotypic areas with strong laminar differentiation. Findings were robust after controlling for spatial autocorrelations using nonparametric spin tests. Mapping connectivity alterations at the edge-level revealed widespread alterations in structural connectivity between individuals with autism and controls, supporting the sensitivity of these analyses to map autism-related network alterations. On the other hand, the microcircuit modeling approach harnessed both structural and functional connectivity information synergistically, making a direct comparison of the relative sensitivity of edge-level comparison vis a vis the modeling approach difficult.

While the former is optimized to detect differences in structural connectivity between groups, the latter reveals additional insights into structure–function coupling and also provides a useful bridge between perturbed macroscale connectivity, on the one hand, and microcircuit dysfunction, on the other.

Spatial association analysis between macroscale manifold distortions and postmortem gene expression maps from the Allen Institute for Brain Sciences (AIBS) pointed to potential neurobiological substrates of our manifold level findings. Recent studies in healthy brain organization[115,116], development[39,65], and disease[117,118] have shown how such analyses can help to understand the relationship between macroscopic neuroimaging phenotypes and spatial variations at the molecular scale[119]. In a prior study, similar approaches were used to identify genetic factors whose expression correlated to maps of cortical morphological variations in autism, and pointed to transcriptionally downregulated genes implicated in autism[120]. In our study, similarly, developmental and disease enrichment analyses exhibited gene sets that were expressed in the cortex and thalamus during childhood and adolescence in autism by associating gene expression patterns with our macroscale findings, suggesting potential interactions between sensory-related subcortical areas and idiotypic/default-mode cortices. The genes that went into the enrichment analysis had strong spatial correlations between reference genes provided by AIBS and manifold findings in the cortical mask, and correlations are thus dependent on cortical differentiation only. However, these genes are not necessarily localized in the cortical areas alone, as the AIBS dataset covers the whole brain[69,121]. Indeed, our results indicated that the cortical areas showing manifold distortions host genes that are also significantly co-expressed in the thalamus. The thalamus relays afferent sensory inputs to the cortex and modulates efferent motor signals. Being a critical hub node in integrative cortico-cortical connectivity in both health and disease[122,123], the thalamus is furthermore recognized to regulate overall levels of cortical excitability[7,11]. In other words, atypical excitatory input from the thalamus would likely lead to altered perceptual input and may thus contribute to sensory abnormalities in autism[11]. Indeed, it has been shown that abnormal functional connectivity between primary sensory cortices and subcortical regions affects the balance between sensory information processing and top-down feedback from higher-order cortices[124]. The connectivity between thalamus and cortical networks extensively modulates brain-wide communication, which means abnormal thalamocortical connectivity likely affects multiple functional processes relevant to autism, including socio-cognitive impairments but also sensory anomalies[7,11,24].

Our multilevel analyses were carried out in a spatially unconstrained manner, yet findings pointed to a co-existence of connectional and microcircuit perturbations in heteromodal association regions (particularly posterior default-mode nodes) and even more strongly to idiotypic/somatomotor cortices. In addition to widely recognized impairments in communication and socio-cognitive functions, individuals with autism show obvious deficits in sensorimotor behaviors[16,114] and these are subsumed under the "repetitive behaviors and interests" syndrome cluster—a core criterion for autism diagnosis. More broadly, autism is increasingly thought to be associated with early sensory anomalies, also formulated in the "sensory-first" hypothesis[16], where atypical formation and maturation of sensory processing circuits in early development may have cascading effects on the development of higher-order networks, which generally mature later and are mediating more integrative and socio-communicative functions[6,8]. Stratifying our cohort into children and adults, we observed more marked sensorimotor network perturbations in the former, while anomalies in

heteromodal association and paralimbic areas were only visible in adults with autism. Similar to recent work from our group that assessed functional hierarchy in autism[6], structural manifold features used in this study were useful in predicting both impairments in lower-level repetitive behavior symptoms as well as higher-order social and cognitive deficits.

Our study provides a potential perspective to consolidate multiple scales of autism pathophysiology. Harnessing advanced connectomics, machine learning, and computational modeling, we could show macroscale structural connectome perturbations in somatosensory/idiotypic and default-mode/heteromodal association areas in autism, which are associated with behavioral symptoms at an individual subject level. These macroscale distortions were also found to relate to cortical microcircuit function in individuals with autism in an in silico model of brain function, in our cohort mostly visible as an increase in excitatory subcortical drive. Despite assessing cross-site variability, implementing bootstrap-based assessment, and running cross-validations where appropriate, our sample size was modest which potentially limits the generalizability of our results. Of note, although we leveraged ABIDE, the currently largest neuroimaging database in autism that is openly accessible, the restriction to individuals who had dMRI, rs-fMRI, and structural MRI data of adequate quality reduced the available sample size. Prior conceptual and simulation findings suggested that a small sample size may yields high variance in predictive errors and emphasized the value of confirming findings in independent and large-scale datasets where possible, not to underestimate prediction error[125]. This can benefit from the aggregation and sharing of more open datasets, ultimately strengthening the generalizability of diagnostic biomarkers[125–127]. A further avenue may also involve the study of transdiagnostic cohorts, which would not only provide additional consolidation of our findings, but also help to evaluate how specific this pattern is to autism, and to further explore inter-individual heterogeneity within the condition[128–131]. Yet, these findings overall provide consistent support that atypical subcortico-cortical interactions, likely between thalamic and sensorimotor areas, contribute to large-scale network anomalies in autism and may suggest that connectivity anomalies of sensorimotor networks that mature early may cascade into overall disorganization of cortico-cortical systems in autism.

## Methods

**Participants**. We studied imaging and phenotypic data of 47 individuals with autism and 37 typically developing controls from the Autism Brain Imaging Data Exchange initiative (ABIDE-II; https://fcon_1000.projects.nitrc.org/indi/abide)[58,59]. Participants were taken from two independent sites: (1) New York University Langone Medical Center (NYU) and (2) Trinity College Dublin (TCD), which were the only sites that included children and adults with autism and neurotypical controls, with ≥10 individuals per group, and who had full MRI data (i.e., structural, functional, and diffusion) available. These 84 participants were selected from a total of 120 participants through the following inclusion criteria: (i) complete multimodal imaging data, i.e., T1-weighted, resting-state functional MRI (rs-fMRI), and dMRI, (ii) acceptable cortical surface extraction, (iii) low head motion in the rs-fMRI time series, i.e., less than 0.3 mm framewise displacement. Individuals with autism were diagnosed by an in-person interview with clinical experts and gold standard diagnostics of Autism Diagnostic Observation Schedule (ADOS)[73] and/or Autism Diagnostic Interview-Revised (ADI-R)[132]. Neurotypical controls did not have any history of mental disorders. For all groups, participants who had genetic disorders associated with autism (i.e., Fragile X), psychological disorders comorbid with autism, contraindications to MRI scanning, and pregnant were excluded. Detailed demographic information of the participants is reported in Supplementary Table 1. The ABIDE data collections were performed in accordance with local Institutional Review Board guidelines. In accordance with HIPAA guidelines and 1000 Functional Connectomes Project/INDI protocols, all ABIDE datasets have been fully anonymized, with no protected health information included.

**MRI acquisition**. At the NYU site, multimodal imaging data were acquired using 3 T Siemens Allegra. T1-weithed data were obtained using a 3D magnetization

prepared rapid acquisition gradient echo (MPRAGE) sequence (repetition time (TR) = 2,530 ms; echo time (TE) = 3.25 ms; inversion time (TI) = 1100 ms; flip angle = 7°; matrix = 256 × 192; and voxel size = 1.3 × 1.0 × 1.3 mm$^3$). The rs-fMRI data were acquired using a 2D echo planar imaging (EPI) sequence (TR = 2000 ms; TE = 15 ms; flip angle = 90°; matrix = 80 × 80; number of volumes = 180; and voxel size = 3.0 × 3.0 × 4.0 mm$^3$). Finally, dMRI data were obtained using a 2D spin echo EPI (SE-EPI) sequence (TR = 5200 ms; TE = 78 ms; matrix = 64 × 64; voxel size = 3 mm$^3$ isotropic; 64 directions; b-value = 1000 s/mm$^2$; and 1 b0 image).

At the TCD site, imaging data were acquired using 3 T Philips Achieva. T1-weighted MRI were acquired using a 3D MPRAGE (TR = 8400 ms; TE = 3.90 ms; TI = 1,150 ms; flip angle = 8°; matrix = 256 × 256; voxel size = 0.9 mm$^3$ isotropic). The rs-fMRI data were aquired using a 2D EPI (TR = 2000 ms; TE = 27 ms; flip angle = 90°; matrix = 80 × 80; number of volumes = 210; and voxel size = 3.0 × 3.0 × 3.2 mm$^3$). Finally, dMRI data were acquired using a 2D SE-EPI (TR = 20,244 ms; TE = 79 ms; matrix = 124 × 124; voxel size = 1.94 × 1.94 × 2 mm$^3$; 61 directions; b-value = 1500 s/mm$^2$; and 1 b0 image).

**Data preprocessing**. T1-weighted data were processed using FreeSurfer[133–138], which includes gradient nonuniformity correction, skull stripping, intensity normalization, and tissue segmentation. White and pial surfaces were generated through triangular surface tessellation, topology correction, inflation, and spherical registration to fsaverage. We obtained preprocessed rs-fMRI data from the ABIDE database (http://preprocessed-connectomes-project.org/abide/), where rs-fMRI data were processed via C-PAC (https://fcp-indi.github.io)[139], including slice timing and head motion correction, skull stripping, and intensity normalization. Nuisance variables of head motion, average white matter and cerebrospinal fluid signal, and linear/quadratic trends were removed using CompCor[140]. Bandpass filtering between 0.01 and 0.1 Hz was applied, and rs-fMRI data were co-registered to T1-weighted data in MNI152 space with boundary-based rigid-body and nonlinear transformations. The rs-fMRI data were mapped to subject-specific midthickness surfaces and resampled to Conte69. Finally, surface-based spatial smoothing with a full-width-at-half-maximum of 5 mm was applied. The dMRI data was processed using MRtrix[30,31], including correction for susceptibility distortions, head motion, and eddy currents. Quality control involved visual inspection of T1-weighted data, and cases with faulty cortical segmentation were excluded. Data with a framewise displacement of rs-fMRI data >0.3 mm were also excluded[141,142].

**Structural connectome manifold identification**. Structural connectomes were generated from preprocessed dMRI data using MRtrix[30,31]. Anatomical constrained tractography was performed using different tissue types derived from the T1-weighted image, including cortical and subcortical grey matter, white matter, and cerebrospinal fluid[33]. The T1-weighted was registered to the dMRI data with boundary-based registration, and the transformation was applied to different tissue types to register them onto the native dMRI space. The multishell and multitissue response functions were estimated[35] and constrained spherical deconvolution and intensity normalization were performed[34]. Seeding from all white matter voxels, the tractogram was generated based on a probabilistic approach[30,31,143] with 40 million streamlines, with a maximum tract length of 250 and a fractional anisotropy cutoff of 0.06. Subsequently, spherical deconvolution informed filtering of tractograms (SIFT2) was applied to optimize an appropriate cross-section multiplier for each streamline[32], and the whole-brain streamlines weighted by the cross-section multipliers are reconstructed. The structural connectome was built by mapping the reconstructed cross-section streamlines onto the Schaefer atlas with 200 parcels[144] then log-transformed[145].

Cortex-wide structural connectome manifolds were identified using BrainSpace (https://github.com/MICA-MNI/BrainSpace)[41]. First, a template manifold was estimated using a group representative structural connectome, defined using a distance-dependent thresholding that preserves long-range connections[60]. The group representative structural connectome was constructed using both autism and control data. A cosine similarity matrix, capturing similarity of connections among different brain regions, was constructed without thresholding the structural connectome, and manifolds were estimated via diffusion map embedding (Fig. 1a, b). Diffusion map embedding is robust to noise and computationally efficient compared to other nonlinear manifold learning techniques[146,147]. It is controlled by two parameters α and t, where α controls the influence of the density of sampling points on the manifold (α = 0, maximal influence; α = 1, no influence) and t controls the scale of eigenvalues of the diffusion operator. We set α = 0.5 and t = 0 to retain the global relations between data points in the embedded space[6,40,41,50,148]. In this new manifold, interconnected brain regions are closely located and regions with weak interconnectivity located farther apart. After generating the template manifold, individual-level manifolds were then estimated and aligned to the template manifold via Procrustes alignment[41,61].

**Between-group differences in structural manifolds**. After controlling for age, sex, and site, multivariate analyses compared individuals with autism and controls in the manifold spanned by the first three structural eigenvectors, which explained more than 50% in structural connectome variance and corresponded to the clearest elbow in the scree plot. We repeated the multivariate analyses with permutation tests by randomly assigning autism and control groups 1000 times. The null

distribution was constructed, and the p-value was calculated by dividing the number of permuted t-statistic values (i.e., Hotelling's T) larger than real t-statistic by the number of permutations. The p-values were FDR-corrected for multiple comparisons[149]. Summary statistics were calculated based on an atlas of laminar differentiation and cortical hierarchy (Fig. 1c)[62] and a widely used community parcellation (Supplementary Fig. 1a)[63]. We furthermore stratified between-group effects for each hemisphere according to cortical hierarchical levels[62] and functional communities[63] (Supplementary Fig. 1b). To simplify the multivariate manifold representations into a single scalar, we quantified manifold eccentricity as the Euclidean distance between the center of template manifold and all data points (i.e., brain regions) in the manifold space for each individual after alignment (Supplementary Fig. 1c)[64,65]. Group averaged manifold eccentricity was compared between individuals with autism and controls to assess the manifold-affected brain regions. It is increasingly recognized that individuals with autism show atypical subcortico-cortical, as well as cerebello-cortical connectivity[150–154]. To assess structural connectivity between subcortical/cerebellar seed regions and cortical targets in autism and controls, we first segmented subcortical regions from T1-weighted data[155] and defined the cerebellum[156]. For each individual, we projected the streamline strength to cortical manifold space by weighting the cortical manifolds with the streamline cross-section of the connection between each subcortical/cerebellar region and cortical parcels to construct weighted manifolds (wM). After controlling for age, sex, and site, we performed 1000 permutation tests with multivariate analyses to compare these dimensions between autism and controls (Supplementary Fig. 7), and FDR-corrected for multiple comparisons[149].

**Microscale neural dynamic modeling.** Large-scale biophysical circuit modeling was conducted to simulate coordinated neuronal activities across the whole brain based on long-range structural connectome information and to estimate regional cellular level parameters of neuronal populations. Specifically, we harnessed a relaxed mean-field model that captures the link between cortical functional dynamics and structural connectivity derived from dMRI, and its modulation through region-specific microcircuit parameters[18]. In comparison to other models that also include synapse-level parameters, this model has a more synoptic scale, allowing for structure–function simulations with modest parametric complexity. For details on the model and its mathematical underpinnings, we refer to the original publication on the relaxed mean-field model[18] and earlier work on the use of (nonrelaxed) mean-field models[42]. In brief, these models approximate the dynamics of spiking and interconnected neural networks through a set of simplified nonlinear stochastic differential equations. Mean-field models assume that neural dynamics of a given region are governed by (i) recurrent intra-regional input, i.e., recurrent excitation/inhibition; (ii) inter-regional input, mediated by dMRI-based structural connections from other nodes, (iii) extrinsic input, mainly from subcortical regions, and (iv) neuronal noise[18]. While the original (non-relaxed) mean-field models[42] assume these parameters to be constant across brain regions, the relaxed mean-field variant allows recurrent excitation/inhibition and subcortical/external input to vary. In the model, global brain dynamics of the network of interconnected local networks is described by the following coupled nonlinear stochastic differential equations[18]:

$$\dot{S}_i = -\frac{S_i}{\tau s} + r(1 - S_i)H(x_i) + \sigma \nu_i(t) \tag{1}$$

$$H(x_i) = \frac{ax_i - b}{1 - \exp(-d(ax_i - b))} \tag{2}$$

$$x_i = WJS_i + GJ \sum_j C_{ij} S_i + I \tag{3}$$

For a given region $i$, $S_i$ in formula (1) represents the average synaptic gating variable, $H(x_i)$ in formula (2) is the population firing rate, and $x_i$ in formula (3) is the total input current. The input current $x_i$ is determined by the recurrent connection strength $W_i$ (i.e., excitation/inhibition) and the excitatory input $I_i$, such as from subcortical relays (i.e., subcortical/external input), and inter-regional signal flow. The latter is governed by $C_{ij}$, which represents the structural connectivity between regions $i$ and $j$, and the global coupling $G$. The global constant $G$ scales the strength of information flow from other cortical regions to the region $i$, relative to the recurrent connection and excitatory inputs. In Eq. (1), the $\nu_i$ term refers to uncorrelated Gaussian noise, modulated by an overall noise amplitude $\sigma$. Following prior work[18], we set parameters as $J = 0.2609$ nA, $a = 270$ n/C, $b = 108$ Hz, $d = 0.154$ s, $r = 0.641$, and $\tau s = 0.1$ s.

We fed the group representative structural connectivity matrix, defined using a distance-dependent thresholding that preserves long-range connections[60], and group averaged functional connectivity matrix into the relaxed mean-field model optimization, which provided recurrent connection strengths $W$ and excitatory subcortical inputs $I$ for every cortical region, as well as a global coupling constant $G$ and a global noise amplitude $\sigma$. During parameter estimation, the simulated synaptic activities $S_i$ are fed into the Balloon–Windkessel hemodynamic model[157] to simulate fMRI signals of each cortical region. The synaptic activity causes an increase in vasodilatory signal $z_i$. Inflow $f_i$ responds in proportion to this signal with concomitant changes in blood volume $\nu_i$ and deoxyhemoglobin content $q_i$.

These biological processes are expressed with following equations[157]:

$$\dot{z}_i = S_i - \kappa z_i - \gamma(f_i - 1) \tag{4}$$

$$\dot{f}_i = z_i \tag{5}$$

$$\tau \dot{\nu}_i = f_i - \nu_i^{1/\alpha} \tag{6}$$

$$\tau \dot{q}_i = \frac{f_i}{\rho}\left[1 - (1 - \rho)^{1/f_i}\right] - q_i \nu_i^{1/\alpha - 1} \tag{7}$$

The parameters were determined by following prior work[157], where resting oxygen extraction fraction $\rho = 0.34$, rate of signal decay $\kappa = 0.65$ s$^{-1}$, rate of elimination $\gamma = 0.41$ s$^{-1}$, hemodynamic transit time $\tau = 0.98$ s, and Grubb's exponent $\alpha = 0.32$. Given $q_i$ and $\nu_i$, the fMRI signal is given as follows[158,159]:

$$\text{fMRI signal}_i = V_0\left[k_1(1 - q_i) + k_2\left(1 - \frac{q_i}{\nu_i}\right) + k_3(1 - \nu_i)\right] \tag{8}$$

The $V_0 = 0.02$ is the resting blood volume fraction and $k_1$, $k_2$, and $k_3$ are a set of parameters dependent of magnetic field strength and a number of acquisition-dependent parameters as follows[158]:

$$k_1 = 4.3\vartheta_0 \rho \text{TE} \tag{9}$$

$$k_2 = \varepsilon r_0 \rho \text{TE} \tag{10}$$

$$k_3 = 1 - \varepsilon \tag{11}$$

The parameter $\vartheta_0 = 28.265 B_0$ is the frequency offset at the outer surface of magnetized vessels and depends on the main magnetic field strength $B_0$, which is 3 T. The $\varepsilon = 0.47$ is the intravascular and extravascular MR signal, and TE is the echo time.

Global and region-specific parameters were determined by maximizing the similarity between simulated and empirical functional connectivity, based on a previously developed neural mass model inversion based on the expectation-maximization algorithm[157,160]. Linear correlations between empirical functional connectivity (FC) and structural connectivity (SC), and that with the simulated functional connectivity (FC') of control data were calculated to assess the quality of the microcircuit parameter estimation (Fig. 2a and S8a). These procedures were performed with a five-fold cross-validation framework with random separation of training and test data, and final microcircuit parameters were determined by averaging across cross-validations. To assess whether the biophysical model predicted FC better than baseline SC, we implemented three different approaches. First, we counted the number of cross-validation folds, where the correlation between FC and FC' was higher than the correlation between SC and FC. Second, we performed 1000 permutation tests by randomly assigning elements of FC' and SC. We calculated the differences in correlations (i.e., corr(pFC', FC) − corr(pSC, FC), where p denotes permutation) 1000 times and constructed a null distribution. If the real difference in correlations (i.e., corr(FC', FC) − corr(SC, FC)) fell outside of the 95% confidence interval of the null distribution, we considered the correlation between empirical and simulated functional connectivity to be significantly higher than the correlation between structural and empirical functional connectivity. Only one side of the null distribution was considered. Finally, we evaluated variation of the relaxed mean-field model using 1000 bootstraps by randomly sampling 90% of subjects with replacement within each group. To reduce the computational complexity, we first reduced the dimensionality of structural and functional connectomes based on seven established functional communities[63]. Then, we repeated estimating microcircuit parameters 1000 times with the dimensionality reduced connectomes (Supplementary Fig. 8b). We calculated correlation coefficients between SC and FC 1000 times and constructed a distribution of the baseline correlations. If the mean correlation between FC and FC' across bootstraps falls outside of the 95% of the baseline distribution, then we considered the model to significantly improve the correspondence between empirical and simulated functional connectivity compared to baseline. Only one side of the null distribution was considered. The robustness of the estimated model parameters was assessed by calculating cross correlations of recurrent excitation/inhibition and subcortical/external input across five folds. The estimated recurrent excitation/inhibition and subcortical/external input parameters were compared between individuals with autism and controls using permutation tests by randomly assigning autism and control groups 1000 times (Supplementary Fig. 8c). The null distribution was constructed, and if the real difference in each parameter between the groups did not belong to 95% of the null distribution, it was deemed significant. The p-values were corrected using FDR[149]. Linear correlations were calculated between the differences in these model-derived parameters between groups and t-statistics of multivariate analysis (Fig. 2b) to evaluate the association between macroscale structural connectome reorganization and imbalances in microcircuit properties. Significances of spatial correlations were assessed via 1000 spin test permutations with randomly rotated microcircuit parameters[66].

The group representative structural connectome that underwent distance-dependent thresholding had a density of 6.31%. A prior study that originally introduced the relaxed mean-field model[18] created the group representative structural connectome by averaging subject-specific matrices, in addition to performing a 50% consistency threshold. When we followed this approach, the

group representative structural connectome would be even sparser (density = 4.20%). We, therefore, assessed consistency of the estimated microcircuit parameters across different structural connectivity matrix thresholds, with decreasing consistency constraints ranging from 50% (4.20% density), 20% (8.78% density), 10% (12.85% density), and 0% (just group average; 35.33% density) of the subjects with streamlines (Supplementary Fig. 9).

**Transcriptomic analysis**. To provide additional neurobiological context for our findings, we assessed spatial correlations between the between-group differences in the structural manifold and gene expression patterns (Fig. 3a). Initially, we correlated the *t*-statistics map derived from the multivariate group comparison and the postmortem gene expression maps provided by Allen Institute for Brain Sciences (AIBS) using the Neurovault gene decoding tool[67,68]. Neurovault implements mixed-effect analysis to estimate associations between the input *t*-statistic map and the genes of AIBS donor brains without considering subcortical regions by masking them out from the input data yielding the gene symbols associated with the input *t*-statistic map. For each gene, a linear model fits the input map to each of the six brains donated to the AIBS. A one-sample *t*-test assessed whether the relation between the gene expression and input *t*-statistic map are consistent across the donated brains. Gene symbols passing FDR-corrected $p < 0.05$ were further validated by assessing whether the same gene list would have been derived from randomly rotated cortical maps. We thus computed a null distribution of spatial correlations between the expression patterns of the identified gene list and 100 randomly rotated maps of the multivariate manifold differences. The actual correlation *t*-statistic was placed into this null distribution to assess significance, and findings were again FDR-corrected. We further examined which of the significant genes were consistently expressed across donors using abagen (https://github.com/rmarkello/abagen)[69]. For each gene, we correlated the whole-brain gene expression map between all pairs of donors, and considered only genes with an average inter-donor r > 0.5 for subsequent analyses. In a second stage, the significant gene list was fed into enrichment analysis (Fig. 3b), which involved comparison against developmental expression profiles from the BrainSpan dataset (http://www.brainspan.org) using the cell-type-specific expression analysis (CSEA) developmental expression tool (http://genetics.wustl.edu/jdlab/csea-tool-2)[53]. As the AIBS repository is composed of adult postmortem datasets, it should be noted that the associated gene symbols represent indirect associations with the input *t*-statistic map derived from the developmental data. To replicate the gene enrichment results with a different database, we additionally performed transcriptomic association analysis using the Genotype-Tissue Expression (GTEx) database (https://www.gtexportal.org/home; Supplementary Fig. 10a). We used the multigene query function, which calculates transcripts per million (TPM) of each gene to quantify the degree of enrichment to a given brain structure. We entered the top 30 ranked genes derived from Neurovault to the multigene query of GTEx. To explore whether the Neurovault derived genetic signature was associated with autism pathophysiology, we additionally performed disease enrichment analysis using previously published transcriptome findings for autism, schizophrenia, and bipolar disorder (Fig. 3c)[70]. A robust linear regression model was constructed for linking the significance of the gene expressions (i.e., t-statistic) derived from Neurovault with log fold-change of autism, schizophrenia, and bipolar disorder, which share similar genetic variants[161]. The fold-change represents the level to which a gene is over or under expressed in a particular condition[70]. Guanine-cytosine (GC) content was controlled to avoid possible effects related to genome size in microarray data[162,163]. To address cell-type-specific gene enrichment, we compared the genes associated with multivariate connectome manifold changes with CSEA proposed in prior work[71,72]. The distinct cell types include excitatory and inhibitory neuronal subtypes in the cortex, and non-neuronal cells of endothelial cells, smooth muscle cells or pericytes, astrocytes, oligodendrocytes and their precursor cells, and microglia[71,72]. For each cell type, we calculated the overlap ratio of how many genes expressed for manifold changes are included in each cell-type-specific genes (Supplementary Fig. 10b). To assess the significance of the overlap ratio, we performed 1000 permutation tests. Among all cell-type-specific genes, we assigned the genes to each cell type with the same gene length. Then, we calculated the overlap ratio between the genes expressed for manifold changes and the permuted cell-type-specific genes. For each cell type, we calculated the overlap ratio 1000 times and constructed a null distribution. If the real overlap ratio did not belong to 95% of the null distribution, it was deemed significant. Multiple comparisons across different cell types were corrected using the FDR procedure[149].

**Symptom severity prediction**. We adopted a supervised machine learning framework with nested cross validation[54–57] to predict autism symptoms measured by ADOS[73]. We aimed at predicting total ADOS scores, as well as subscores for social cognition, communication, and repeated behavior/interest (Fig. 4 and Supplementary Table 3). We utilized five-fold nested cross validation[54–57] and elastic net regularization[74] with regularization parameters ranging from 0.1 (i.e., more to L2-norm) to 1.0 (i.e., L1-norm). Nested cross-validation split the dataset into training (4/5) and test (1/5) partitions, and each training partition was further split into inner training and testing folds using another five-fold cross validation. The model

with lowest overfitting across the inner folds was applied to the test partition of the outer fold. After controlling for age, sex, and site from a total of 600 (200 regions × 3 manifolds) features, we selected performant features using elastic net regularization. ADOS score prediction leveraged linear regression with the selected features. The procedure was repeated 100 times with different training and test partitions. Prediction accuracy was benchmarked with linear correlations between the actual and predicted ADOS scores and the mean absolute error (MAE), and their 95% confidence interval. Permutation-based correlations across 1000 tests were conducted by randomly shuffling ADOS scores to verify whether the prediction performance exceeded chance levels. We repeated the prediction analysis using three-fold nested cross validation, so that each fold includes more training and test data (Supplementary Table 4).

**Sensitivity and specificity analyses.**

(a) Site effects: The multivariate group comparison using structural connectome manifolds was performed for each site (NYU and TCD separately) to see the consistency of results across different sites (Supplementary Fig. 1d). We also repeated the symptom severity prediction for each site using five-fold nested cross validation to assess whether the prediction results are consistent across different sites (Supplementary Table 5 and 6).

(b) Head motion effects: To rule out whether the macroscale perturbations in autism related to head motions, we first calculated mean FD from dMRI for all participants (Supplementary Fig. 2a). Two-sample *t*-test assessed between-group differences in head motion. In addition, we repeated the multivariate manifold comparisons between groups while controlling for mean FD (Supplementary Fig. 2b).

(c) Age effects: To assess the age-related effects on structural connectome manifolds, we performed multivariate group comparison in manifolds, controlled for sex and site, within children (age < 18) and adults (age ≥ 18) cohorts separately (Supplementary Fig. 3).

(d) Associations to cortical morphology: Several studies have previously reported atypical cortical morphology in individuals with autism relative to controls[12,19]. To assess whether these morphological variations contribute to our connectome results, we calculated linear correlations between multivariate findings and cortical morphology measures (i.e., cortical thickness and cortical curvature) between groups (Supplementary Fig. 4a). We also repeated the multivariate manifold comparisons while controlling for cortical thickness and curvature to evaluate whether the connectome-wide effects can be observed above and beyond potential variations in cortical morphology (Supplementary Fig. 4b).

(e) Analysis of each manifold dimensions: We compared each manifold dimension between individuals with autism and controls (Supplementary Fig. 5). Between-group differences were assessed using 1000 permutation tests. We FDR-corrected for multiple comparisons[149].

(f) Inter-regional connection effects: We also compared the edge values of the structural connectivity matrix (i.e., streamline cross-section) between individuals with autism and controls (Supplementary Fig. 6). The significance of the between-group difference was assessed using 1000 permutation tests followed by FDR-correction[149]. We further performed ADOS score prediction using streamline cross-section with five-fold nested cross validation (Supplementary Table 7)[54–57].

**Reporting summary**. Further information on research design is available in the Nature Research Reporting Summary linked to this article.

## Data availability

The imaging and phenotypic data were provided, in part, by the Autism Brain Imaging Data Exchange initiative (ABIDE-II; https://fcon_1000.projects.nitrc.org/indi/abide/)[59]. Data for transcriptomic analysis were obtained from BrainSpan dataset (http://www.brainspan.org) and Genotype-Tissue Expression (GTEx) database (https://www.gtexportal.org/home). The subsets of data from these databases that were used in the present work are available from the authors upon request. Source data are provided with this paper.

## Code availability

The codes for connectome manifold generation are available at https://github.com/MICA-MNI/BrainSpace[41]; codes for computational circuit modeling are available at https://github.com/ThomasYeoLab/CBIG/tree/master/stable_projects/fMRI_dynamics/Wang2018_MFMem[18]. Transcriptomic association analyses were conducted using NeuroVault (https://neurovault.org), cell-type-specific expression analysis (CSEA) (http://genetics.wustl.edu/jdlab/csea-tool-2)[53], and abagen tools (https://github.com/rmarkello/abagen)[69].

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

## Acknowledgements

B.P. was funded by the National Research Foundation of Korea (NRF-2020R1A6A3A03037088), a Molson Neuro-Engineering fellowship by Montreal Neurological Institute and Hospital (MNI), and Fonds de la Recherche du Québec—Santé (FRQ-S). C.P. was funded through a postdoctoral fellowship of the FRQ-S. O.B. was funded by a Healthy Brains for Healthy Lives (HBHL) postdoctoral fellowship. R.A.I.B. was funded by a British Academy Post-Doctoral Fellowship and the Autism Research Trust. A.G. was funded by the Simons Foundation (SFARI 400101), Brain and Behavior Foundation (NARSAD - National Alliance for Research on Schizophrenia and Depression), the European Research Council (ERC - DISCONN, GA802371), the NIH (1R21MH116473-01A1) and the Telethon foundation (GGP19177). B.T.T.Y. was supported by the Singapore NRF Fellowship (Class of 2017) and the National University of Singapore Yong Loo Lin School of Medicine (NUHSRO/2020/124/TMR/LOA). Any opinions, findings and conclusions or recommendations expressed in this material are those of the authors and do not reflect the views of National Research Foundation, Singapore. J.S. was supported by the European Research Council (WANDERINGMINDS-ERC646927). B.C.B. acknowledges research support from the National Science and Engineering Research Council of Canada (NSERC Discovery-1304413), the Canadian Institutes of Health Research (CIHR FDN-154298), SickKids Foundation (NI17-039), Azrieli Center for Autism Research (ACAR-TACC), BrainCanada (Azrieli Future Leaders), FRQ-S, and the Tier-2 Canada Research Chairs program. B.P., C.P., R.A.I.B., and B.C.B. are jointly funded through an MNI-Cambridge collaborative award. The Genotype-Tissue Expression (GTEx) Project was supported by the Common Fund of the Office of the Director of the National Institutes of Health, and by NCI, NHGRI, NHLBI, NIDA, NIMH, and NINDS.

## Author contributions

B.P. and B.C.B. designed the experiments, analyzed the data, and wrote the manuscript. S.H., C.P., and O.B. aided with the experiments. S.V., R.A.I.B., A.D.M., M.P.M., A.G., B.T.T.Y., and J.S. reviewed the manuscript. B.P. and B.C.B. are the corresponding authors of this work and have responsibility for the integrity of the data analysis.

## Competing interests

The authors declare no competing interests.
