## [Peer Review File · Nature Communications]

Reviewer #1 (Remarks to the Author):

This is a novel and interesting paper that utilizes cutting-edge manifold learning methods to investigate structural brain connectivity in individuals with autism. The biophysical model and transcriptomic analyses are key strengths. The paper is timely and provides new insights into the architecture of the structural connectome in autism. Given the current sample size and significant phenotypic heterogeneity associated with autism, future studies may need to assess the reproducibility of these findings in an independent cohort and characterize heterogeneity in individual-specific manifolds.

1. The manifold distortions and between-group differences in micro-circuit parameters ultimately boil down to differences in interregional streamline counts. It would be informative to investigate and localize potential between-group differences in the structural connectivity matrices. This could reveal specific connections (i.e. white matter tracts) that underlie the manifold distortions.

2. It seems that further work would be required to establish the robustness and reliability of the estimated micro-circuit parameters. Importantly, confidence intervals should be established for the parameter estimates and statistical inference should be performed to determine whether the between-group differences (Fig. 2c) exceed uncertainty and degeneracy in the model fitting process. Bootstrapping could be used if the model fitting process is computationally efficient. Alternatively, assessing variation in parameter fits across the five folds may be helpful in this regard, although further folds may be needed. The concern here is that the model is over-parameterized and thus a degenerate solution space could lead to unstable parameter fits (i.e. there may be multiple solutions that yield comparable model fits). Can it be shown that the optimal solution is unique? Ideally, the model could be fitted to each individual connectome, avoiding the need for any bootstrapping on the group-level parameter estimates.

3. The accuracy of the biophysical model might be queried. The correlation between predicted and empirical functional connectivity is marginally higher than the correlation between empirical structural and functional connectivity (0.26 and 0.23, respectively). As such, it is unclear whether the model enhances predictive utility beyond structural connectivity. Is the model largely recapitulating structural connectivity? Consideration might be given to partialling out the effect of structural connectivity when assessing the correlation between predicted and empirical functional connectivity.

4. Were manifold distortions investigated for subcortical regions and the cerebellum? Previous work highlights the importance of the insula (Francis et al, 2018), cingulate cortex and thalamus in the disorder. Cerebellar regions have also implicated in autism (Stoodley et al, 2017; Ramos et al, 2018). In the present study, manifold distortions tend to be greatest in temporal and somatomotor cortices of the right hemisphere (Fig. 1C). Perhaps testing each manifold separately might provide further insight, such as overlapping distortions across the three manifolds. Any insight into the strong lateralization of the distortion?

5. The t-statistic map (Fig. 1C) appears to be continuously distributed across the cortical surface, yet inference was presumably undertaken at scale of atlas regions. Given that the test statistic utilized requires estimation of sample covariances and the modest sample size, it would be prudent to estimate p-values with permutation-based inference and it may be informative to perform inference on each manifold separately.

6. The associations between symptom severity and the structural manifolds is very interesting and future work would need to establish whether these predictions hold in an independent cohort. As a benchmark, it would be informative to evaluate whether structural connectivity (i.e. streamline counts) achieve comparable prediction accuracies. Does the probability map in Fig. 4 account for regional consistency across the three manifolds? It is not clear why spin tests are required to assess significance. Presumably randomizing the correspondence between individuals and manifold maps would be sufficient to establish the null.

7. Further details about the connectome mapping pipeline are required, including tractography algorithm and streamline seeding methodology. The structural connectome in Fig. 2A appears to

be quite heavily thresholded, although the details of the thresholding procedure are scant. How was the threshold selected? If probabilistic tractography was used, some form of thresholding can improve reconstruction accuracy (Sarwar et al, 2018) and suppress false positives. This could potentially improve the fit of the biophysical model. Using SIFT in case-control comparisons can potentially lead to paradoxical changes and some caution may be warranted here.

In summary, this study is highly novel and provides new systems-level insight into structural brain connectivity differences associated autism. Importantly, the biophysical model provides specific insight into potential processes underlying these distortions, including excitation-inhibition and subcortical inputs. Inclusion of transcriptomic analyses enhances innovation. Comprehensive and very timely work. Establishing robustness of the parameter fits from the biophysical model and explicitly testing for between-group differences in these estimates would be helpful in the revision.
Signed, Andrew Zalesky

Reviewer #2 (Remarks to the Author):

The manuscript "Connectome and microcircuit models implicate atypical subcortico-cortical interactions in autism pathophysiology" presents a beautifully executed, sophisticated, analysis of structural and functional connectomics, in combination with neural mass modelling and transcriptomics to present and integrated understanding of autism spectrum disorder. The results are well integrated with current theoretical perspectives such as the "sensory-first" hypotheses of autism.

A have some general technical comments.

a) The methods specify that a published parcellation (Schaefer et al 2018 - 200 parcel version) was used throughout all structural and functional connectome analyses. However, brain maps of the results (especially figure 1C) are blurred at the edges between parcels – which makes it appear to the reader that the analysis was completed vertex-wise and produced large significant clusters. Please consider revising to plots with demarcated parcel borders to clarify that this is a parcellated analysis.

b) Please provide more detail about how the fMRI and DWI data were corrected for susceptibility distortions and registered to the T1w anatomical data within the preprocessing pipelines. Was boundary-based registration (BBR) employed? Was diffusion tractography seeded from the white matter surface?

c) I find the figure legend for Figure 2b confusing. What does the hexagon size reflect? Why does it not scale with color if it is the p-value. Are there two different types of p-value? Please consider expanding the figure caption for clarity.

d) Of greater concern, the gene list produced from the AHBA analysis is heavily enriched for cortex specific genes. When the gene list is uploaded to <https://hbasel.msl.ubc.ca/>, it is very clear that this gene list is comprised of genes that differentiate the cerebral cortex from other (subcortical) regions. The authors state that this genes list came from a Neurovault gene decoding tool. Does this tools mask out subcortical regions when they to omit them from the transcriptomic analysis? Or does it take the absence of subcortical data in the t-maps as evidence negative weights in these areas? Would the same gene list have been derived from the null (i.e. rotated) cortical maps used for a statistical control in the other sections of this manuscript are uploaded to Neurovault? The interpretation of the rest of this section (i.e. the developmental and the disease enrichment analysis) is strongly hindered if this is only a "cortex-only" gene list – with no differentiation across the cortex.

e) While the machine- learning analysis is well performed, with an appropriate nested 5-fold cross-validation scheme. Given the known heterogeneity between autism studies (including the heterogeneity reported in this papers supplemental analysis according to recruiting site, and age), as well as, the limited sample size (only 47 individuals with autism – meaning less than 10

participants per fold), I feel that any analysis that claims to predict individual symptom scores feels too good to be true and should be interpreted with extreme caution until sufficiently large independent replication samples can also be analyzed. The authors may consider "downplaying" these results in light of the small sample size at this time.

f) It is surprising that the interpretation is very focused on subcortical-cortical interactions but MR based measures of subcortical-cortical connectivity (for example thalamocortical structural or functional connectivity) from this sample are not analyzed in this paper. Do the inferred excitatory subcortical inputs from the mean-field model correspond to measured case-control differences in thalamocortical connectivity?

g) Consider revising for grammar: "However, the connectome-wide manifold distortions were found to correlate with increases in excitatory subcortical drive into cortical microcircuits level, especially in lower-level cortical hierarchical areas, even after controlling for spatial autocorrelations using non-parametric spin tests, and to marginally relate to increases in excitation and inhibition."

Reviewer #3 (Remarks to the Author):

In this manuscript, the authors perform extensive analyses of connectome microcircuit dysfunction in autism, using manifold learning, a computational model for functional connectivity, gene enrichment analysis and a symptom prediction model on a small subset of subjects from the ABIDE sample.

Whilst I found this paper interesting, I have several critical problems with it as it stands. First, the authors have crammed so many analyses into one paper that it makes it hard to parse and especially difficult to assess the work properly. The methods description of each step is very brief and in many cases quite unclear. In my opinion there is not sufficient detail provided to understand and replicate the analysis steps taken. I will provide a non-exhaustive list below:

- it is totally unclear to me what the authors refer to by their "multivariate analysis". Are they testing coefficients derived from the manifold learning step? If so, which ones?
- The authors mention in passing a "seminal model of neural organization" (p5) but do not explain this at all. For example, what is idiotypic cortex?
- The microscale neural dynamic model is not explained well at all. Please provide considerably more detail on how the model was estimated, what the parameters refer to, how they were estimated etc. I would also like to see sensitivity analysis to see how robust the choice of these parameters.
- the motivation and implementation for the spin test for clinical score prediction is also unclear. Why have the authors chosen to do this via the alternative of (e.g.) shuffling the labels? Also, procedurally please provide more information on how this was done.

Another major concern that I have relates to the possibility of overfitting. The analysis pipelines are all very complex and the authors have a tiny sample size (47 individuals with autism and 37 controls) which is well known to lead to substantial variation in accuracy (e.g. for a decoding analysis such as the authors pursue here see e.g. Varoquaux NIMG 2018). As far as I can see this could be a possibility for at least two analyses:

1. For the cross-validation of parameters for microscale neural dynamic modelling, the authors use 5 fold cross validation, but not nested cross-validation. The final microcircuit parameters were obtained by averaging across folds. This will almost certainly provide positively biased estimates of predictive performance because a strict training test separation has not been preserved. A related problem is that this hides variation in the parameter estimates due to perturbations due to the cross-validation. I would like to see how variable the estimates are across folds. As noted above,

given the sample size I would expect this to be substantial.

2. The symptom prediction (ADOS scores) uses an elastic net. The authors use a fixed regularization parameter setting but again I suspect this could be biased because the authors again do not mention nested cross-validation and again select features that were repeatedly selected across cross-validation folds for the final predictive model.

From a conceptual level how do the authors reconcile their manifold learning approach with a parcellation approach that follows? The results are not necessarily complementary with one another since the manifold learning approach estimates multiple *spatially overlapping* patterns of connectivity. The parcellation approach that follows necessarily involves averaging these signals into piece-wise constant parcels which potentially mixes signals from the large scale cortical gradients. See Haak et al NIMG 2018, 2020 for further discussion on this issue.

Other minor points:

- I am not convinced the authors have done enough to address site effects. For example, I would also like to see symptom prediction models estimated separately for each site (after addressing the concerns about overfitting above)

- the authors do not do any statistical tests to determine whether the dynamical model for connectivity predicted functional connectivity better than the baseline model. Given the small sample size, I doubt whether a difference of $r = 0.26$ on the test set is really different from 0.23 (see p5)

Reviewer #4 (Remarks to the Author):

The paper by Park et al. carries out connectome study comparing autism cases and neurotypical controls and then relates the connectome changes to gene expression data from the Allen brain atlas. The transcriptome analysis is potentially interesting, but it needs additional supporting evidence and validation in an independent gene expression dataset.

Major points:

1. It is not clear how the gene decoding analysis was done and therefore whether the approach is appropriate. Was the t-statistics map correlated with gene expression data from each individual from the Allen Brain atlas, or was gene expression averaged across individuals? If averaged, how is inter-individual variation taken into account statistically?
2. Are the observed correlations replicable in an independent dataset (eg. GTEX)?
3. The authors need to address the possibility that the observed correlations may be driven by cell type composition differences between brain regions. Are the significantly correlated genes enriched for cell-type specific genes?

REVIEWER 1:

This is a novel and interesting paper that utilizes cutting-edge manifold learning methods to investigate structural brain connectivity in individuals with autism. The biophysical model and transcriptomic analyses are key strengths. The paper is timely and provides new insights into the architecture of the structural connectome in autism. Given the current sample size and significant phenotypic heterogeneity associated with autism, future studies may need to assess the reproducibility of these findings in an independent cohort and characterize heterogeneity in individual-specific manifolds.

We thank the Reviewer for the positive evaluation and helpful comments, which we addressed point-by-point below.

1. The manifold distortions and between-group differences in micro-circuit parameters ultimately boil down to differences in interregional streamline counts. It would be informative to investigate and localize potential between-group differences in the structural connectivity matrices. This could reveal specific connections (i.e., white matter tracts) that underlie the manifold distortions.

As suggested, we compared the edge values of the structural connectivity matrix (i.e., streamline cross-section) between individuals with autism and controls. As also suggested by the Reviewer in #5, comparisons followed a permutation approach. Specifically, after regressing out age, sex, and site from each element of the structural connectivity matrix, we compared the residuals between individuals with autism and controls. We repeated this process by randomly assigning autism and control groups 1,000 times. The null distribution was constructed, and the p-value was calculated by dividing the number of permuted t-statistics that exceeded the actual t-value by the number of permutations. The p-values were corrected using FDR to adjust for multiple comparisons. Among the 19,900 connections (i.e., the upper triangular elements in the connectivity matrix), 623 edges showed significant differences (Fig. S6A; 481/142 increased/decreased in individuals with autism). We charted the t-statistics of these significantly different connections with circular plots (Fig. S6B). Interestingly, we found the most marked (i.e., top 5%) increases took place mainly for short range tracts (average streamline length 9.37 mm), while most marked decreases involved longer range connections (20.90 mm, $p < 0.001$, Fig. S6C).

Fig. S6 | Edge-wise differences in streamline cross-section between typically developing controls and individuals with autism. (A) The t-statistics of the whole connections (lower triangular) showed significant between-group differences (upper triangular) in streamline cross-section between individuals with autism and controls. Findings have been corrected for multiple comparisons at false discovery rate (FDR) < 0.05 . **(B)** Circular plots represent the connections that showed significant between-group differences. Increased/decreased streamline cross-sections in individuals with autism are represented in red/blue. All

significant connections are reported with high transparency and top 50/25/10% t-statistic values are reported with less transparency for better visualization. The solid lines indicate connections with top 5% t-statistic values. (C) Average streamline distance for top 5% increased and decreased connections. Error bars indicate standard error of the mean.

We updated the *Results* (P.5):

“To explore specific connections that potentially contribute to the above manifold distortions, we compared the streamline cross-section between individuals with autism and controls. We could observe a mix of connectivity increases as well as decreases in autism relative to controls, affecting multiple networks, including those in which we observed manifold anomalies (Fig. S6A–B). Notably, lengths of fiber tracts showing decreases were considerably longer (20.90 mm) compared to fiber tracts showing increases (9.37 mm) (Fig. S6C).”

as well as the *Methods* (P.19):

“f) Inter-regional connection effects. We also compared the edge values of the structural connectivity matrix (i.e., streamline cross-section) between individuals with autism and controls (Fig. S6). Significance of the between-group difference was assessed using 1,000 permutation tests followed by FDR-correction¹³⁷.”

2. It seems that further work would be required to establish the robustness and reliability of the estimated micro-circuit parameters. Importantly, confidence intervals should be established for the parameter estimates and statistical inference should be performed to determine whether the between-group differences (Fig. 2c) exceed uncertainty and degeneracy in the model fitting process. Bootstrapping could be used if the model fitting process is computationally efficient. Alternatively, assessing variation in parameter fits across the five folds may be helpful in this regard, although further folds may be needed. The concern here is that the model is over-parameterized, and thus a degenerate solution space could lead to unstable parameter fits (i.e., there may be multiple solutions that yield comparable model fits). Can it be shown that the optimal solution is unique? Ideally, the model could be fitted to each individual connectome, avoiding the need for any bootstrapping on the group-level parameter estimates.

We thank the Reviewer for suggesting additional tests to establish robustness of the biophysical network model. When we performed the five-fold cross-validation with different training and test datasets, mean \pm SD of the estimated microcircuit parameters across the different cross-validation folds were 0.530 ± 0.004 for recurrent excitation/inhibition and 0.325 ± 0.001 for subcortical/external input in controls. Similarly, consistent findings were observed in individuals with autism, with 0.528 ± 0.006 for recurrent excitation/inhibition and 0.325 ± 0.001 for subcortical/external input across folds.

As suggested, we also evaluated model parameter variation using bootstraps within each group (1,000 bootstraps, randomly subsampling 90% of subjects with replacement). As the model is computationally prohibitive for a large number of regions (i.e., in the case of 200×200 , a single iteration takes ~ 30 h on a 15-core UNIX computer with 252 GB RAM), we first reduced the dimensionality of structural and functional connectomes based on seven established functional communities to make this analysis tractable⁶². For each functional network, we normalized the microcircuit parameters of controls to mean zero and SD of one, and those of individuals with autism were normalized according to controls. The mean \pm SD of the microcircuit parameters across 1,000 bootstraps of individuals with autism relative to controls are reported as radar plots in the **Fig. S8C** below:

Fig. S8 | Microcircuit parameters and biophysical simulations. (C) Microcircuit parameters of controls and individuals with autism. The functional community-wise⁶² stratification of the parameters are presented with radar plots. Black lines indicate controls normalized to mean zero and standard deviation of one, and red lines indicate individuals with autism normalized according to controls. Solid and dash lines represent mean and standard deviation of the parameters across 1,000 bootstraps, respectively.

We updated the *Results* (P.6):

“Estimated microcircuit parameters were consistent for the cross-validation (Fig. 2B and S8C), with a mean \pm SD across the five folds of 0.530 ± 0.004 for recurrent excitation/inhibition and 0.325 ± 0.001 for subcortical/external input in controls, and 0.528 ± 0.006 for recurrent excitation/inhibition and 0.325 ± 0.001 for subcortical/external input in autism. To confirm stability, we ran a bootstrap-based evaluation of the relaxed mean-field model based on an intrinsic functional community partitioning⁶² (see Methods). The mean \pm SD of the parameters across 1,000 bootstraps in autism relative to controls were consistent (Fig. S8C).”

as well as the *Methods* (P.16):

“We additionally evaluated variation of the relaxed mean-field model using 1,000 bootstraps by randomly sampling 90% of subjects with replacement within each group. To reduce the computational complexity, we first reduced the dimensionality of structural and functional connectomes based on seven established functional communities⁶². Then, we repeated estimating microcircuit parameters 1,000 times with the dimensionality reduced connectomes (Fig. S8B).”

The between-group differences in microcircuit parameters between individuals with autism and controls were assessed using permutation tests by randomly assigning autism and control groups 1,000 times. The null distribution was constructed, and if the real difference in each parameter between groups did not belong to 95% of the null distribution, it was deemed significant. The p-values across functional communities were corrected using FDR. We found significant increases in recurrent excitation/inhibition in visual ($p = 0.02$) and limbic networks ($p < 0.001$) in autism, while subcortical/external inputs decreased in dorsal attention ($p < 0.001$), frontoparietal ($p = 0.02$), and default mode networks ($p = 0.01$), as well as increase in sensorimotor network ($p < 0.001$) (**Fig. S8C**).

We updated the *Results*: (P.6):

“Between-group differences in microcircuit parameters between autism and controls were assessed using permutation tests with 1,000 iterations. We found increased recurrent excitation/inhibition in visual ($p = 0.02$) and limbic networks ($p < 0.001$) in autism; considering subcortical/external inputs, we observed decreases in dorsal attention ($p < 0.001$), frontoparietal ($p = 0.02$), and default mode networks ($p = 0.01$), while sensorimotor networks increased ($p < 0.001$) (Fig. S8C).”

as well as the *Methods* (P.17):

“The estimated recurrent excitation/inhibition and subcortical/external input parameters were compared between individuals with autism and controls using permutation tests by randomly assigning autism and control groups 1,000 times (Fig. S8C).”

The null distribution was constructed, and if the real difference in each parameter between the groups did not belong to 95% of the null distribution, it was deemed significant. The p -values were corrected using FDR¹³⁷.

3. The accuracy of the biophysical model might be queried. The correlation between predicted and empirical functional connectivity is marginally higher than the correlation between empirical structural and functional connectivity (0.26 and 0.23, respectively). As such, it is unclear whether the model enhances predictive utility beyond structural connectivity. Is the model largely recapitulating structural connectivity? Consideration might be given to partialling out the effect of structural connectivity when assessing the correlation between predicted and empirical functional connectivity.

At a high resolution, the biophysical model achieved a correlation between empirical and simulated functional connectivity of $r = 0.50$ for the training and of 0.26 for the test data, which was higher than the baseline correlation between structural connectivity and empirical functional connectivity ($r = 0.25$ for training, $r = 0.23$ for test). Due to the computational complexity (see *above*), we could not assess statistical significance for this improvement in predicting functional connectivity with the biophysical model when the 200×200 data were used. Instead, we applied the same dimensionality reduction procedure and bootstraps as described above. Here, we calculated correlation coefficients between structural connectivity and empirical functional connectivity 1,000 times to construct a baseline null distribution. If the mean correlation between empirical and simulated functional connectivity across bootstraps did not belong to 95% of the baseline null distribution, the model was interpreted to significantly improve the correspondence between empirical and simulated functional connectivity compared to baseline. Only one side of the null distribution was considered. For controls, the optimal model indeed predicted functional connectivity ($r = 0.68$ for training, $r = 0.65$ for test), outperforming baseline correlations between structural and functional connectivity ($r = 0.53$, $p < 0.001$ for training, $r = 0.49$, $p < 0.001$ for test) (**Fig. S8B**). Significant improvements were also observed in individuals with autism (**Fig. S8B**; $p < 0.001$ for both training and test), indicating that the biophysical model significantly improved the association between structural and functional connectivity.

Fig. S8 | Microcircuit parameters and biophysical simulations. (B) Linear correlations between empirical functional connectivity (FC) and structural connectivity (SC), and empirical and simulated FC for controls and individuals with autism based on functional communities⁶². Black lines indicate mean correlation and gray lines represent 95% confidence interval across 1,000 bootstrapping.

We updated the *Results* (P.6):

“We assessed statistical significance for the improvement in predicting of functional connectivity with the biophysical model compared to a baseline model based on structural connectivity by conducting bootstraps using dimensionality reduced structural and functional connectomes based on functional communities⁶² (see *Methods*). The correlation coefficients between empirical and simulated functional connectivity outperformed the corresponding baseline correlations between structural and functional connectivity in both controls and individuals with autism (*Fig. S8B*; $p < 0.001$ for both training and test).”

as well as the *Methods* (P.16-17):

“We also calculated correlation coefficients between structural connectivity and empirical functional connectivity 1,000 times and constructed a null distribution. If the mean correlation between empirical and simulated functional connectivity across bootstraps did not belong to 95% of the baseline null distribution, we considered the model to significantly improve the correspondence between empirical and simulated functional connectivity compared to baseline. Only one side of the null distribution was considered.”

4. Were manifold distortions investigated for subcortical regions and the cerebellum? Previous work highlights the importance of the insula (Francis et al., 2018), cingulate cortex and thalamus in the disorder. Cerebellar regions have also implicated in autism (Stoodley et al., 2017; Ramos et al., 2018). In the present study, manifold distortions tend to be greatest in temporal and somatomotor cortices of the right hemisphere (Fig. 1C). Perhaps testing each manifold separately might provide further insight, such as overlapping distortions across the three manifolds. Any insight into the strong lateralization of the distortion?

We thank the Reviewer for pointing this out. To assess subcortical and cerebellar effects in individuals with autism, we first estimated streamline cross-section from each subcortical and cerebellar region to cortical areas. Here, we segmented subcortical regions from T1-weighted data¹⁴³ and defined the cerebellum using the atlas provided by FSL¹⁴⁴. For each individual, we projected the streamline strength to the cortical manifold space by weighting the cortical manifolds with the streamline cross-section of the connection between each subcortical/cerebellar node and cortical parcels to construct weighted manifolds (wM). For each subcortical and cerebellar structure, multivariate analysis compared weighted manifolds spanned by wM1–wM3 between individuals with autism and controls while controlling for age, sex, and site. We performed permutation tests, which randomly re-assigned autism and control groups 1,000 times as described above (#1), and FDR-corrected the resultant p-values. We found significant between-group differences in subcortical (or cerebellar)-weighted manifolds for thalamus ($p = 0.03$), caudate ($p = 0.04$), and cerebellum ($p = 0.03$), indicating the low dimensional representation of subcortico-cortical connectivity in these regions changed in autism (Fig. S7).

Fig. S7 | Subcortical and cerebellar manifold distortions in autism. Cortical manifolds weighted by streamline cross-section of individuals with autism (red) and controls (blue) for each subcortical region and cerebellum. Each transparent dot represents each individual and solid dots indicate the center of the weighted manifolds of each group. The t-statistics of the multivariate between-group comparison in weighted manifolds (wM1, wM2, and wM3) and significance corrected for multiple comparisons at false discovery rate (FDR) < 0.05 are reported.

We updated the *Results* (P.5):

“To further assess subcortico-cortical and cerebello-cortical connectivity, we projected the streamline strength of these regions to cortical targets in each individual to the manifold space (see *Methods*). For each subcortical and cerebellar

structure, multivariate analysis compared weighted manifolds (wM) spanned by wM1–wM3 between individuals with autism and controls while controlling for age, sex, and site. After FDR-correction, we found significant between-group differences in the thalamus (FDR = 0.03), caudate (FDR = 0.04), and cerebellum (FDR = 0.03), indicating the low dimensional representation of subcortico-cortical connectivity in these regions changed in autism (Fig. S7).”

as well as the *Methods* (P. 15)

“It is increasingly recognized that individuals with autism show atypical subcortico-cortical, as well as cerebello-cortical connectivity^{138–142}. To assess structural connectivity between subcortical/cerebellar seed regions and cortical targets in autism and controls, we first segmented subcortical regions from T1-weighted data¹⁴³ and defined the cerebellum¹⁴⁴. For each individual, we projected the streamline strength to cortical manifold space by weighting the cortical manifolds with the streamline cross-section of the connection between each subcortical/cerebellar region and cortical parcels to construct weighted manifolds (wM). After controlling for age, sex, and site, we performed 1,000 permutation tests with multivariate analyses to compare these dimensions between autism and controls (Fig. S7), and FDR-corrected for multiple comparisons¹³⁷.”

As also suggested, we furthermore evaluated between-group differences for each manifold dimension separately. The first dimension showed significant effects in lateral temporal regions and the second and third dimensions showed significant effects in medial temporal and lateral somatosensory areas (Fig. S5). Overall, patterns were similar to the multivariate findings, but effects were stronger when considering all dimensions simultaneously.

Fig. S5 | Distortions in each structural connectome manifold. The t-statistics of the identified regions that showed significant between-group differences in each dimension between individuals with autism and controls. Stratification of between-group differences effects along cortical hierarchical levels (*i.e.*, middle column)⁶¹ and functional network⁶² are presented in the radar plots.

We updated the *Results* (P. 5):

“We also compared each manifold dimension between individuals with autism and controls. We found that the first dimension showed significant effects in lateral temporal regions, while the second and third dimensions showed significant effects in medial temporal and lateral somatosensory areas (Fig. S5). Patterns were similar to the multivariate findings, but effects were stronger when considering all dimensions simultaneously.”

as well as the *Methods* (P.19):

“e) Analysis of each manifold dimensions. We compared each manifold dimension between individuals with autism and controls (Fig. S5). Between-group differences were assessed using 1,000 permutation tests. We FDR-corrected for multiple comparisons¹³⁷.”

The Reviewer also asked regarding the lateralization of connectome manifold distortions. While a systematic evaluation of inter-hemispheric differences was beyond the scope of this work, we stratified between-group manifold difference effects according to cortical hierarchical levels⁶¹, as well as functional communities⁶², for left and right hemispheres separately. Considering hierarchical levels, effects were % stronger in the right hemisphere in idiotypic and somatomotor network, while they were % stronger in the left hemisphere for heteromodal association cortices. Effects were very similar in left and right paralimbic and unimodal areas. These findings are now presented in (Fig. S1B).

Fig. S1 | Connectome manifold distortions. (B) The t-statistics of between-group differences in the whole-brain. Stratification of between-group differences effects for each hemisphere along cortical hierarchical levels (*i.e.*, middle column) ⁶¹ and functional network ⁶² are presented in the radar plots.

We updated the *Results* (P.5):

“To address the lateralization of findings, we stratified the between-group effects according to cortical hierarchical levels ⁶¹ for left and right hemispheres separately (Fig. S1B). We found X% stronger effects in right compared to left idiotypic networks and X% stronger effects in left vs right networks. A comparable effect was seen when stratifying findings across intrinsic functional communities ⁶²”

the *Discussion* (P.11):

“Notably, connectome manifold distortions showed more marked effects in the right hemisphere for idiotypic areas, while they showed more marked effects in left heteromodal areas. Several prior studies suggested asymmetric findings in autism at the level of cortical morphology as well as functional organization ^{88 86,87,91 85,90 89}. Our results may contribute to these prior findings by showing a potentially differential susceptibility of the left and right hemisphere for lower vs higher level structural reorganization in autism.

and the *Methods* (P.15):

“We furthermore stratified between-group effects for each hemisphere according to cortical hierarchical levels ⁶¹ and functional communities ⁶² (Fig. S1B).”

5. The t-statistic map (Fig. 1C) appears to be continuously distributed across the cortical surface, yet inference was presumably undertaken at scale of atlas regions. Given that the test statistic utilized requires estimation of sample covariances and the modest sample size, it would be prudent to estimate p-values with permutation-based inference and it may be informative to perform inference on each manifold separately.

As suggested, we implemented permutation tests to evaluate between-group differences in connectome manifolds. After regressing out age, sex, and site from the connectome manifolds, we performed multivariate analysis to compare residuals between autism and controls. We repeated this process by randomly assigning autism and control groups 1,000 times. The null distribution was constructed, and the p-value was calculated by dividing the number of permuted t-statistic values larger than real t-value with the number of permutations. The p-values of all brain regions were FDR-corrected for multiple comparisons. This approach identified a slightly more restricted, yet overall similar topography of manifold changes. We replaced the original approach with the permutation test, and revised **Fig. 1C**:

Fig. 1 | Structural connectome manifolds. (C) The t-statistics of the identified regions that showed significant between-group differences in these dimensions between individuals with autism and controls. Findings have been corrected for multiple

comparisons at false discovery rate (FDR) < 0.05. Stratification of between-group differences effects along cortical hierarchical levels (*i.e.*, middle column)⁶¹ is presented in the radar plots.

We updated the *Methods* (P.14):

*“After controlling for age, sex, and site, multivariate analyses compared individuals with autism and controls in the manifold spanned by the first three structural eigenvectors, which explained more than 50% in structural connectome variance and corresponded to the clearest elbow in the scree plot. We repeated the multivariate analyses with permutation tests by randomly assigning autism and control groups 1,000 times. The null distribution was constructed, and the p-value was calculated by dividing the number of permuted t-statistic values (*i.e.*, Hotelling’s T) larger than real t-statistic by the number of permutations. The p-values were FDR-corrected for multiple comparisons¹³⁷.”*

We also changed relevant Supplementary Figures using the permutation-based approach (**Fig. S1, S2, S3, and S4**). As the identified regions have changed (*i.e.*, *Fig. 1C*), we changed **Fig. 2B** and **Fig. 3** accordingly. In all cases, results were highly consistent with our original findings.

We also evaluated between-group differences for each manifold dimension separately (see #4, above).

6. The associations between symptom severity and the structural manifolds is very interesting and future work would need to establish whether these predictions hold in an independent cohort. As a benchmark, it would be informative to evaluate whether structural connectivity (*i.e.*, streamline counts) achieve comparable prediction accuracies. Does the probability map in Fig. 4 account for regional consistency across the three manifolds? It is not clear why spin tests are required to assess significance. Presumably randomizing the correspondence between individuals and manifold maps would be sufficient to establish the null.

As suggested, we also performed a symptom severity prediction using the edge values of the structural connectivity matrix (*i.e.*, streamline cross-section) based on five-fold nested cross-validation^{54–57}. Similar to the results from connectome manifolds, regularization parameters of 0.6 and 0.7 showed good performance for predicting ADOS scores, where the selected connections were primarily found in medial prefrontal, lateral parietal, lateral temporal, posterior cingulate, and sensorimotor areas (**Table S7**). Prediction significance was indeed established by permutation tests that randomly shuffled ADOS scores, and not on spin tests. We apologize for the typo.

We updated the *Results* (P.10):

*“Lastly, we performed symptom severity prediction using the edge values of the structural connectivity matrix (*i.e.*, streamline cross-section). Similar to the results from connectome manifolds, regularization parameter of 0.6 and 0.7 showed good performance for predicting ADOS scores, where the selected connections were primarily found in medial prefrontal, lateral parietal lobule, lateral temporal, posterior cingulate, and sensorimotor regions (**Table S7**).”*

as well as the *Methods* (P.18, 19):

“Permutation-based correlations across 1,000 tests were conducted by randomly shuffling ADOS scores to verify whether the prediction performance exceeded chance levels.”

*“f) Inter-regional connection effects. We further performed ADOS score prediction using streamline cross-section with five-fold nested cross-validation (**Table S7**)^{54–57}.”*

7. Further details about the connectome mapping pipeline are required, including tractography algorithm and streamline seeding methodology. The structural connectome in Fig. 2A appears to be quite heavily thresholded, although the details of the thresholding procedure are scant. How was the threshold selected? If probabilistic tractography was used, some form of thresholding can improve reconstruction accuracy (Sarwar et al, 2018) and suppress false positives. This could potentially

improve the fit of the biophysical model. Using SIFT in case-control comparisons can potentially lead to paradoxical changes and some caution may be warranted here.

We are happy to further clarify our approach and also explore the use of thresholding. First, we used probabilistic tractography^{30,31,132} together with SIFT2 tractogram filtering³² for connectome generation and have now specified this further in the *Methods*. In the probabilistic approach, a streamline is more probable to follow the predominant fiber orientation density, but it also rarely traverses paths where the orientation density is relatively small^{30,31,132}. The SIFT2 algorithm optimizes an appropriate cross-section multiplier for each streamline to match a whole-brain tractogram to voxel-wise fiber densities, reducing false positives³². We updated the *Methods* for clarification (P.14):

“Seeding from all white matter voxels, the tractogram was generated based on a probabilistic approach^{30,31,132} with 40 million streamlines, with a maximum tract length of 250 and a fractional anisotropy cutoff of 0.06. Subsequently, spherical-deconvolution informed filtering of tractograms (SIFT2) was applied to optimize an appropriate cross-section multiplier for each streamline³², and the whole-brain streamlines weighted by the cross-section multipliers are reconstructed.”

We did not threshold the individual-level structural connectivity matrices, as they were already sparse (mean density across subjects = $6.81 \pm 1.69\%$). For the relaxed mean-field model, we fed in a group representative structural connectome that underwent distance-dependent thresholding that preserves long-range connections⁶⁰. This group representative matrix has connection density of 6.31%. We added the information to the *Methods* (P.16):

“We fed the group representative structural connectivity matrix, defined using a distance-dependent thresholding that preserves long-range connections⁶⁰, and group averaged functional connectivity matrix into the relaxed mean-field model optimization, ~”

A prior study originally introducing the relaxed mean field model¹⁸ created the group representative structural connectome by averaging subject specific matrices, in addition to performing a 50% consistency threshold. When we followed this approach, the group representative structural connectome would be even sparser (density = 4.20%). We therefore assessed consistency of the estimated microcircuit parameters across different structural connectivity matrix thresholds, with decreasing consistency constraints ranging from 50% (4.20% density), 20% (8.78% density), 10% (12.85% density), and 0% (just group average; 35.33% density) of the subjects with streamlines. The estimated microcircuit parameters were largely consistent across different thresholds except for recurrent excitation/inhibition in autism (**Fig. S9**). Notably, between-group differences in parameters did not show significant differences across different thresholding measures, indicating our approach is reasonable.

Fig. S9 | Microcircuit parameters estimated using different connectome thresholds. The recurrent excitation/inhibition and subcortical/external input with distant-dependent thresholding⁶⁰ and consistency of 50, 20, 10, and 0% threshold¹⁸ of

each group, as well as between-group differences, across five-fold cross-validations are reported. Significant differences in parameters across different thresholds are reported with asterisk.

We updated the *Results* (P.6):

“Estimated microcircuit parameters were largely consistent across different matrix thresholds (Fig. S9). Notably, between-group differences in parameters did not show significant differences across different thresholding procedures, suggesting the sensitivity of our approach.”

as well as the *Methods* (P.17):

“The group representative structural connectome that underwent distance-dependent thresholding had a density of 6.31%. A prior study that originally introduced the relaxed mean field model¹⁸ created the group representative structural connectome by averaging subject specific matrices, in addition to performing a 50% consistency threshold. When we followed this approach, the group representative structural connectome would be even sparser (density = 4.20%). We therefore assessed consistency of the estimated microcircuit parameters across different structural connectivity matrix thresholds, with decreasing consistency constraints ranging from 50% (4.20% density), 20% (8.78% density), 10% (12.85% density), and 0% (just group average; 35.33% density) of the subjects with streamlines (Fig. S9).”

In summary, this study is highly novel and provides new systems-level insight into structural brain connectivity differences associated autism. Importantly, the biophysical model provides specific insight into potential processes underlying these distortions, including excitation-inhibition and subcortical inputs. Inclusion of transcriptomic analyses enhances innovation. Comprehensive and very timely work. Establishing robustness of the parameter fits from the biophysical model and explicitly testing for between-group differences in these estimates would be helpful in the revision.

Signed, Andrew Zalesky

We would like to thank Dr Zalesky for the positive evaluation and helpful comments, which we feel have significantly improved our work.

REVIEWER 2:

The manuscript “Connectome and microcircuit models implicate atypical subcortico-cortical interactions in autism pathophysiology” presents a beautifully executed, sophisticated, analysis of structural and functional connectomics, in combination with neural mass modelling and transcriptomics to present and integrated understanding of autism spectrum disorder. The results are well integrated with current theoretical perspectives such as the “sensory-first” hypotheses of autism.

We thank the Reviewer for positive evaluation and constructive comments.

1. I have some general technical comments. a) The methods specify that a published parcellation (Schaefer et al 2018 - 200 parcel version) was used throughout all structural and functional connectome analyses. However, brain maps of the results (especially figure 1C) are blurred at the edges between parcels – which makes it appear to the reader that the analysis was completed vertex-wise and produced large significant clusters. Please consider revising to plots with demarcated parcel borders to clarify that this is a parcellated analysis.

As suggested, we modified all figures to more clearly show parcel data.

2. b) Please provide more detail about how the fMRI and DWI data were corrected for susceptibility distortions and registered to the T1w anatomical data within the preprocessing pipelines. Was boundary-based registration (BBR) employed? Was diffusion tractography seeded from the white matter surface?

We apologize for the omission and have now added more details to the *Methods* (P.13, 14):

“We obtained preprocessed rs-fMRI data from the ABIDE database (<http://preprocessed-connectomes-project.org/abide/>), where rs-fMRI data were processed via C-PAC (<https://fcp-indi.github.io>)¹²⁸, including slice timing and head motion correction, skull stripping, and intensity normalization. Nuisance variables of head motion, average white matter and cerebrospinal fluid signal, and linear/quadratic trends were removed using CompCor¹²⁹. Band-pass filtering between 0.01 and 0.1 Hz was applied, and rs-fMRI data were co-registered to T1-weighted data in MNI152 space with boundary-based rigid-body and non-linear transformations. The rs-fMRI data were mapped to subject-specific midthickness surfaces and resampled to Conte69. Finally, surface-based spatial smoothing with a full-width-at-half-maximum of 5 mm was applied. The dMRI data was processed using MRtrix^{30,31} including correction for susceptibility distortions, head motion, and eddy currents.”

“Structural connectomes were generated from preprocessed dMRI data using MRtrix^{30,31}. Anatomical constrained tractography was performed using different tissue types derived from the T1-weighted image, including cortical and subcortical grey matter, white matter, and cerebrospinal fluid³³. The T1-weighted was registered to the dMRI data with boundary-based registration and the transformation was applied to different tissue types to register them onto the native dMRI space. The multi-shell and multi-tissue response functions were estimated³⁵ and constrained spherical deconvolution and intensity normalization were performed³⁴. Seeding from all white matter voxels, the tractogram was generated based on a probabilistic approach^{30,31,132} with 40 million streamlines, with a maximum tract length of 250 and a fractional anisotropy cutoff of 0.06. Subsequently, spherical-deconvolution informed filtering of tractograms (SIFT2) was applied to optimize an appropriate cross-section multiplier for each streamline³², and the whole-brain streamlines weighted by the cross-section multipliers are reconstructed.”

3. c) I find the figure legend for Figure 2b confusing. What does the hexagon size reflect? Why does it not scale with color if it is the p-value. Are there two different types of p-value? Please consider expanding the figure caption for clarity.

We assume the Reviewer referred to Fig. 3B that contains the hexagons. Here, hexagon sizes do not represent p-values, but the proportion of genes specifically expressed in particular tissue at particular developmental stage. Varying stringency for enrichment with respect to specificity index threshold (pSI) are represented by the size of hexagons going from least specific (outer hexagons) to most specific (center hexagons) with different level of pSI (0.05, 0.01, 0.001, and 0.0001)⁵³. The p-values are relevant to the colors inside the hexagons. We revised the caption of Fig. 3B for clarity.

Fig. 3 | Transcriptomic analysis to identify gene expression patterns. (B) Developmental enrichment, showing strong associations with cortex and thalamus during early childhood and adolescence, as well as early infancy and young adulthood. The size of hexagon rings represents the proportion of genes specifically expressed in particular tissue at particular developmental stage. Varying stringency for enrichment with respect to specificity index threshold (pSI) are represented by the size of hexagons going from least specific (outer hexagons) to most specific (center hexagons) (pSI = 0.05, 0.01, 0.001, and 0.0001, respectively)⁵³. Colors represent the false discovery rate (FDR)-corrected p-values. The bar plot on the left represents the log transformed p-values, averaged across all brain structures.

4. d) Of greater concern, the gene list produced from the AHBA analysis is heavily enriched for cortex specific genes. When the gene list is uploaded to <https://hbaset.msl.ubc.ca/>, it is very clear that this gene list is comprised of genes that differentiate the cerebral cortex from other (subcortical)

regions. The authors state that this genes list came from a Neurovault gene decoding tool. Does this tool mask out subcortical regions when they to omit them from the transcriptomic analysis? Or does it take the absence of subcortical data in the t-maps as evidence negative weights in these areas? Would the same gene list have been derived from the null (i.e. rotated) cortical maps used for a statistical control in the other sections of this manuscript are uploaded to Neurovault? The interpretation of the rest of this section (i.e. the developmental and the disease enrichment analysis) is strongly hindered if this is only a “cortex-only” gene list – with no differentiation across the cortex.

We thank the Reviewer for these thoughtful comments, which we are happy to address below. To first clarify the approach, we associated between-group differences in the cortex-derived structural manifold findings with gene expression patterns in cortical areas only. Neurovault thus excluded subcortical regions (mask shown below). Unlike an approach that does not exclude subcortical data when building the correlations, the masking avoids correlating absent subcortical imaging data with subcortical gene expression patterns that would lead to the artificial effects the Reviewer may be concerned about.

Response letter Fig. 1 | Subcortical mask used for transcriptomic association analysis.

This is now clarified in the *Methods* (P.17):

“Neurovault implements mixed-effect analysis to estimate associations between the input t-statistic map and the genes of AIBS donor brains without considering subcortical regions by masking them out from the input data yielding the gene symbols associated with the input t-statistic map.”

We would then like to better outline how we obtained thalamus-enriched genes. Specifically, the genes going into the enrichment analysis were based on spatial correlations between reference genes and imaging data in cortical areas. These findings are thus dependent on the cortical differentiation of these genes. However, as the AIBS dataset covers the whole brain^{67,112}, the identified genes are not necessarily specific to cortical areas alone. The results indicate that the cortical areas showing manifold distortions host genes that are also significantly co-expressed in the thalamus. We thus updated the *Discussion* section (P.12):

“The genes that went into the enrichment analysis had strong spatial correlations between reference genes provided by Allen Institute for Brain Sciences (AIBS) and manifold findings in the cortical mask, and correlations are thus dependent on cortical differentiation only. However, these genes are not necessarily localized in the cortical areas alone, as the AIBS dataset covers the whole-brain^{67,112}. As such, our results indicate that the cortical areas showing manifold distortions host genes that are also significantly co-expressed in the thalamus.”

We followed the Reviewer’s excellent suggestion and assessed spatial correlations between the expression patterns of the identified gene list and rotated maps of the multivariate manifold differences (100 spherical rotations). For each gene, we thus built a null distribution using the correlation t-statistics into which the actual correlation t-statistic was placed. Actual correlations of all genes reported exceeded rotation-derived null correlations, and findings were significant at $FDR < 0.05$. This control analysis is now added to the *Results* and *Methods* (P.7, 17).

“For significant gene lists after multiple comparisons correction ($FDR < 0.05$), we repeated the transcriptomic association analysis with rotated maps of multivariate change pattern 100 times to ensure that genes were not selected by chance.”

“Gene symbols passing FDR-corrected $p < 0.05$ were further validated, by assessing whether the same gene list would have been derived from randomly rotated cortical maps. We thus computed a null distribution of spatial correlations between the

expression patterns of the identified gene list and 100 randomly rotated maps of the multivariate manifold differences. The actual correlation *t*-statistic was placed into this null distribution to assess significance, and findings were again FDR-corrected.”

5. e) While the machine-learning analysis is well performed, with an appropriate nested 5-fold cross-validation scheme. Given the known heterogeneity between autism studies (including the heterogeneity reported in this papers supplemental analysis according to recruiting site, and age), as well as, the limited sample size (only 47 individuals with autism – meaning less than 10 participants per fold), I feel that any analysis that claims to predict individual symptom scores feels too good to be true and should be interpreted with extreme caution until sufficiently large independent replication samples can also be analyzed. The authors may consider “downplaying” these results in light of the small sample size at this time.

We thank the Reviewer for the positive appraisal of our approach and agree with the helpful suggestion. Firstly, we repeated the symptom severity prediction using three-fold nested cross-validation to include more samples per fold, and found consistent results (**Table S4**). These findings are now presented in the *Results* and *Methods* as follows (P. 10, 18):

“Performing the same analyses using three-fold nested cross-validation, we found largely consistent prediction results. Indeed, the regularization parameter of 0.7 and 0.8 showed good performance for predicting ADOS scores, and features were primarily selected in lateral and medial prefrontal, lateral parietal, and lateral temporal regions (Table S4).”

“We repeated the prediction analysis using three-fold nested cross-validation, so that each fold includes more training and test data (Table S4).”

While these findings further support our approach, we nevertheless agree with the Reviewer’s important recommendation to downplay the interpretation of the predictive results in the *Discussion* (P.12):

“Arguably, the modest sample size of this study poses some limitations on the results from the symptom severity prediction, despite the use of three- and five-fold cross-validations. It, overall, warrants replication of our findings in larger, and ideally transdiagnostic cohorts to also evaluate specificity and to further explore heterogeneity within the autism spectrum itself¹¹⁷⁻¹²⁰.”

6. f) It is surprising that the interpretation is very focused on subcortical-cortical interactions, but MR based measures of subcortical-cortical connectivity (for example thalamocortical structural or functional connectivity) from this sample are not analyzed in this paper. Do the inferred excitatory subcortical inputs from the mean-field model correspond to measured case-control differences in thalamocortical connectivity?

We thank the Reviewer for pointing this out. To examine interactions between subcortical and cortical regions, we conducted additional analyses of subcortico-cortical connectivity. To assess subcortical and cerebellar effects in individuals with autism, we first estimated streamline cross-section from each subcortical and cerebellar region to cortical areas. Here, we segmented subcortical regions from T1-weighted data¹⁴³ and defined the cerebellum using the atlas provided by FSL¹⁴⁴. For each individual, we projected the streamline strength to the cortical manifold space by weighting the cortical manifolds with the streamline cross-section of the connection between each subcortical/cerebellar node and cortical parcels to construct weighted manifolds (wM). For each subcortical and cerebellar structure, multivariate analysis compared weighted manifolds spanned by wM1–wM3 between individuals with autism and controls while controlling for age, sex, and site. We performed permutation tests, which randomly re-assigned autism and control groups 1,000 times as described above (#1), and FDR-corrected the resultant *p*-values. We found significant between-group differences in subcortical (or cerebellar)-weighted manifolds for thalamus (*p* = 0.03), caudate (*p* = 0.04), and cerebellum (*p* = 0.03), indicating the low dimensional representation of subcortico-cortical connectivity in these regions changed in autism (**Fig. S7**).

Fig. S7 | Subcortical and cerebellar manifold distortions in autism. Cortical manifolds weighted by streamline cross-section of individuals with autism (red) and controls (blue) for each subcortical region and cerebellum. Each transparent dot represents each individual and solid dots indicate the center of the weighted manifolds of each group. The t-statistics of the multivariate between-group comparison in weighted manifolds (wM1, wM2, and wM3) and significance corrected for multiple comparisons at false discovery rate (FDR) < 0.05 are reported.

We updated the *Results* (P.5):

“To further assess subcortico-cortical and cerebello-cortical connectivity, we projected the streamline strength of these regions to cortical targets in each individual to the manifold space (see Methods). For each subcortical and cerebellar structure, multivariate analysis compared weighted manifolds (wM) spanned by wM1–wM3 between individuals with autism and controls while controlling for age, sex, and site. After FDR-correction, we found significant between-group differences in the thalamus (FDR = 0.03), caudate (FDR = 0.04), and cerebellum (FDR = 0.03), indicating the low dimensional representation of subcortico-cortical connectivity in these regions changed in autism (Fig. S7).”

as well as the *Methods* (P. 15)

“It is increasingly recognized that individuals with autism show atypical subcortico-cortical, as well as cerebello-cortical connectivity^{138–142}. To assess structural connectivity between subcortical/cerebellar seed regions and cortical targets in autism and controls, we first segmented subcortical regions from T1-weighted data¹⁴³ and defined the cerebellum¹⁴⁴. For each individual, we projected the streamline strength to cortical manifold space by weighting the cortical manifolds with the streamline cross-section of the connection between each subcortical/cerebellar region and cortical parcels to construct weighted manifolds (wM). After controlling for age, sex, and site, we performed 1,000 permutation tests with multivariate analyses to compare these dimensions between autism and controls (Fig. S7), and FDR-corrected for multiple comparisons¹³⁷.”

7. g) Consider revising for grammar: *“However, the connectome-wide manifold distortions were found to correlate with increases in excitatory subcortical drive into cortical microcircuits level, especially in lower-level cortical hierarchical areas, even after controlling for spatial autocorrelations using non-parametric spin tests, and to marginally relate to increases in excitation and inhibition.”*

We thank the Reviewer for pointing this out, and revised the sentence as follows (P.11):

“However, the connectome-wide manifold distortions were found to correlate with increased recurrent excitation/inhibition and excitatory subcortical drive into cortical microcircuits. This was especially marked in idiotypic areas with strong laminar differentiation, even after controlling for spatial autocorrelations using non-parametric spin tests.”

REVIEWER 3:

In this manuscript, the authors perform extensive analyses of connectome microcircuit dysfunction in autism, using manifold learning, a computational model for functional connectivity, gene enrichment analysis and a symptom prediction model on a small subset of subjects from the ABIDE

sample. Whilst I found this paper interesting, I have several critical problems with it as it stands. First, the authors have crammed so many analyses into one paper that it makes it hard to parse and especially difficult to assess the work properly. The methods description of each step is very brief and in many cases quite unclear. In my opinion there is not sufficient detail provided to understand and replicate the analysis steps taken. I will provide a non-exhaustive list below:

We thank the Reviewer for the positive evaluation and for the constructive comments. Please see the point-by-point responses and clarifications below.

1. It is totally unclear to me what the authors refer to by their “multivariate analysis”. Are they testing coefficients derived from the manifold learning step? If so, which ones?

The multivariate analysis compared three connectome manifold dimensions (*i.e.*, a set of eigenvectors) between individuals with autism and controls. We utilized the SurfStat toolbox (<http://www.math.mcgill.ca/keith/surfstat/>) to fit linear models of the formula $b_0 + b_1 \cdot \text{Age} + b_2 \cdot \text{Sex} + b_3 \cdot \text{Site} + b_4 \cdot \text{Group}$, to multivariate data Y (of the form number of subjects \times number of brain regions \times number of manifolds). Here, we compared three structural eigenvectors, which explained 50.6% in structural connectome variance and corresponded to the clearest elbow in the scree plot). Inference is based on Hotelling’s t-test in each parcel. We clarified this in the *Methods* (P.14):

*“After controlling for age, sex, and site, multivariate analyses compared individuals with autism and controls in the manifold spanned by the first three structural eigenvectors, which explained more than 50% in structural connectome variance and corresponded to the clearest elbow in the scree plot. We repeated the multivariate analyses with permutation tests by randomly assigning autism and control groups 1,000 times. The null distribution was constructed, and the p-value was calculated by dividing the number of permuted t-statistic values (*i.e.*, Hotelling’s T) larger than real t-statistic by the number of permutations. The p-values were FDR-corrected for multiple comparisons¹³⁷.”*

2. The authors mention in passing a “seminal model of neural organization” (p5) but do not explain this at all. For example, what is idiotypic cortex?

We are happy to provide further context; the text refers to the model from Marcel Mesulam, 2000, which posits a gradient of laminar and functional differentiation from sensory/motor areas towards transmodal/paralimbic systems along four concentric rings. Level 1 are referred to as idiotypic areas, primary sensory and motor cortices that have a dominant layer 4. Level 2 comprises unimodal association areas (superior and inferior temporal, superior and inferior parietal, and premotor cortices). Level 3 comprises heteromodal association areas (higher-order prefrontal, posterior parietal, lateral temporal, and parahippocampal cortices). Level 4 contains paralimbic cortices (orbitofrontal, insular, temporopolar, and cingulate cortex). The location of each area is represented in Fig. 1C and details can be found in Mesulam et al., 2000. We also provide these details in a new **Table S2**.

Table S2 | Seminal model of neural organization that contains four cortical hierarchy levels.

Cortical hierarchy	Mesulam 2000 ⁶¹	Brodmann 1909 ¹⁵⁰
Idiotypic	Striate	17
	Auditory	41, 42
	Somatosensory	3a, 3b, 1, 2
	Motor	4, 6
Unimodal association	Upstream peristriate	18, 19
	Inferotemporal	20, 21, 37
	Superior temporal	22
	Superior parietal lobule	5, anterior 7
	Inferior parietal lobule	anterior 40
Premotor	anterior 6, posterior 8, 44	
Heteromodal association	Prefrontal cortex	9, 10, 45, 46, 47, anterior 11, anterior 12, anterior 8
	Posterior parietal	posterior 7, 39, 40
	Lateral temporal	parts of 21 & 37

	Parahippocampal	parts of 36 & 37
Paralimbic	Orbitofrontal cortex	posterior 11, posterior 12, 13
	Insula	14, 15, 16
	Temporal pole	38
	Parahippocampal	27, 28, 35
	Cingulate	23, 24, 25, 26, 29, 30, 31, 32, 33

Adapted from Paquola et al., 2019⁴⁰.

3. The microscale neural dynamic model is not explained well at all. Please provide considerably more detail on how the model was estimated, what the parameters refer to, how they were estimated etc. I would also like to see sensitivity analysis to see how robust the choice of these parameters.

As suggested, we revised the *Methods* for more detail and explanation (P.15-16):

“Large-scale biophysical circuit modeling was conducted to simulate coordinated neuronal activities across the whole brain based on long-range structural connectome information and to estimate regional cellular level parameters of neuronal populations. Specifically, we harnessed a relaxed mean-field model that captures the link between cortical functional dynamics and structural connectivity derived from dMRI, and its modulation through region-specific microcircuit parameters¹⁸. In comparison to other models that also include synapse-level parameters, this model has a more synaptic scale, allowing for structure-function simulations with modest parametric complexity. For details on the model and its mathematical underpinnings, we refer to the original publication on the relaxed mean-field model¹⁸ and earlier work on the use of (non-relaxed) mean-field models⁴². In brief, these models approximate the dynamics of spiking and interconnected neural networks through a set of simplified nonlinear stochastic differential equations. Mean-field models assume that neural dynamics of a given region are governed by (i) recurrent intra-regional input i.e., recurrent excitation/inhibition; (ii) inter-regional input, mediated by dMRI-based structural connections from other nodes, (iii) extrinsic input, mainly from subcortical regions, and (iv) neuronal noise¹⁸. While the original (non-relaxed) mean-field models⁴² assume these parameters to be constant across brain regions, the relaxed mean-field variant allows recurrent excitation/inhibition and subcortical/external input to vary. In the model, global brain dynamics of the network of interconnected local networks is described by the following coupled nonlinear stochastic differential equations¹⁸:

$$\dot{S}_i = -\frac{S_i}{\tau_s} + r(1 - S_i)H(x_i) + \sigma v_i(t) \quad (1)$$

$$H(x_i) = \frac{ax_i - b}{1 - \exp(-d(ax_i - b))} \quad (2)$$

$$x_i = WJS_i + GJ \sum_j C_{ij}S_j + I \quad (3)$$

For a given region i , S_i in formula (1) represents the average synaptic gating variable, $H(x_i)$ in formula (2) is the population firing rate, and x_i in formula (3) is the total input current. The input current x_i is determined by the recurrent connection strength W_i (i.e., excitation/inhibition) and the excitatory input I_i , such as from subcortical relays (i.e., subcortical/external input), and inter-regional signal flow. The latter is governed by C_{ij} , which represents the structural connectivity between regions i and j , and the global coupling G . The global constant G scales the strength of information flow from other cortical regions to the region i , relative to the recurrent connection and excitatory inputs. In equation (1), the v_i term refers to uncorrelated Gaussian noise, modulated by an overall noise amplitude σ . Following prior work¹⁸, we set parameters as $J = 0.2609$ nA, $a = 270$ n/C, $b = 108$ Hz, $d = 0.154$ s, $r = 0.641$, and $\tau_s = 0.1$ s.

We fed the group representative structural connectivity matrix, defined using a distance-dependent thresholding that preserves long-range connections⁶⁰, and group averaged functional connectivity matrix into the relaxed mean-field model optimization, which provided recurrent connection strengths W and excitatory subcortical inputs I for every cortical region, as well as a global coupling constant G and a global noise amplitude σ . During parameter estimation, the simulated synaptic activities S_i are fed into the Balloon–Windkessel hemodynamic model¹⁴³ to simulate fMRI signals of each cortical region. The synaptic activity causes an increase in vasodilatory signal z_i . Inflow f_i responds in proportion to this signal with concomitant changes in blood volume v_i and deoxyhemoglobin content q_i . These biological processes are expressed with following equations¹⁴³:

$$\dot{z}_i = S_i - \kappa z_i - \gamma(f_i - 1) \quad (4)$$

$$\dot{f}_i = z_i \quad (5)$$

$$\tau \dot{v}_i = f_i - v_i^{1/\alpha} \quad (6)$$

$$\tau \dot{q}_i = \frac{f_i}{\rho} [1 - (1 - \rho)^{1/f_i}] - q_i v_i^{1/\alpha - 1} \quad (7)$$

The parameters were determined by following prior work¹⁴³, where resting oxygen extraction fraction $\rho = 0.34$, rate of signal decay $\kappa = 0.65 \text{ s}^{-1}$, rate of elimination $\gamma = 0.41 \text{ s}^{-1}$, hemodynamic transit time $\tau = 0.98 \text{ s}$, and Grubb's exponent $\alpha = 0.32$. Given q_i and v_i , the fMRI signal is given as follows^{144,145}:

$$fMRI \text{ signal}_i = V_0 \left[k_1(1 - q_i) + k_2 \left(1 - \frac{q_i}{v_i} \right) + k_3(1 - v_i) \right] \quad (8)$$

The $V_0 = 0.02$ is the resting blood volume fraction and k_1 , k_2 , and k_3 are a set of parameters dependent of magnetic field strength and a number of acquisition-dependent parameters as follows¹⁴⁴:

$$k_1 = 4.3\vartheta_0\rho TE \quad (9)$$

$$k_2 = \varepsilon r_0\rho TE \quad (10)$$

$$k_3 = 1 - \varepsilon \quad (11)$$

The parameter $\vartheta_0 = 28.265B_0$ is the frequency offset at the outer surface of magnetized vessels and depends on the main magnetic field strength B_0 , which is 3T, The $\varepsilon = 0.47$ is the intravascular and extravascular MR signal, and TE is the echo time.

Global and region-specific parameters were determined by maximizing the similarity between simulated and empirical functional connectivity, based on a previously developed neural mass model inversion based on the expectation-maximization algorithm^{143,146}.

In addition, we expanded the *Results* (P.5-6):

“Biophysical computational simulations¹⁸ complemented our macroscale findings by modelling atypical microcircuit-level functional dynamics in autism. Harnessing a relaxed mean-field model¹⁸, we simulated dynamics of functional signals through a set of simplified nonlinear stochastic differential equations by linking ensembles of local neural masses (i.e., theoretical cell population models for excitatory neurons which reciprocally inhibit each other; Fig. 2A) with diffusion-derived structural connectivity, allowing for simulating whole-brain functional connectivity. Notably, the model iteratively tunes its parameters to generate maximally similar functional connectivity patterns compared to empirical data, and it also resulted in an optimal set of biophysical parameters (specifically, recurrent excitation/inhibition and excitatory subcortical/external input; see *Methods*).”

To detail model stability, we now provide mean \pm SD of the parameters across the folds of the cross-validation, which supports high stability. In addition, we evaluated model parameter variation using bootstraps within each group (1,000 bootstraps, randomly subsampling 90% of subjects with replacement). As the model is computationally prohibitive for a large number of regions (i.e., in the case of 200×200 , a single iteration takes $\sim 30\text{h}$ on a 15-core UNIX computer with 252 GB RAM), we first reduced the dimensionality of structural and functional connectomes based on seven established functional communities to make this analysis tractable⁶². For each functional network, we normalized the microcircuit parameters of controls to mean zero and SD of one, and those of individuals with autism were normalized according to controls. The mean \pm SD of the microcircuit parameters across 1,000 bootstraps of individuals with autism relative to controls are reported as radar plots in the **Fig. S8C** below:

Fig. S8 | Microcircuit parameters and biophysical simulations. (C) Microcircuit parameters of controls and individuals with autism. The functional community-wise⁶² stratification of the parameters are presented with radar plots. Black lines indicate controls normalized to mean zero and standard deviation of one, and red lines indicate individuals with autism normalized according to controls. Solid and dash lines represent mean and standard deviation of the parameters across 1,000 bootstraps, respectively.

We updated the *Results* (P.6):

“Estimated microcircuit parameters were consistent for the cross-validation (Fig. 2B and S8C), with a mean \pm SD across the five folds of 0.530 ± 0.004 for recurrent excitation/inhibition and 0.325 ± 0.001 for subcortical/external input in controls, and 0.528 ± 0.006 for recurrent excitation/inhibition and 0.325 ± 0.001 for subcortical/external input in autism. To confirm stability, we ran a bootstrap-based evaluation of the relaxed mean-field model based on an intrinsic functional community partitioning⁶² (see Methods). The mean \pm SD of the parameters across 1,000 bootstraps in autism relative to controls were consistent (Fig. S8C).”

as well as the *Methods* (P.16):

“We additionally evaluated variation of the relaxed mean-field model using 1,000 bootstraps by randomly sampling 90% of subjects with replacement within each group. To reduce the computational complexity, we first reduced the dimensionality of structural and functional connectomes based on seven established functional communities⁶². Then, we repeated estimating microcircuit parameters 1,000 times with the dimensionality reduced connectomes (Fig. S8B).”

4. The motivation and implementation for the spin test for clinical score prediction is also unclear. Why have the authors chosen to do this via the alternative of (e.g.) shuffling the labels? Also, procedurally please provide more information on how this was done.

We apologize for the typo. Prediction significance was indeed established by permutation tests that randomly shuffled ADOS scores, and not on spin tests. This is now revised in the *Methods* (P.18):

“Permutation-based correlations across 1,000 tests were conducted by randomly shuffling ADOS scores to verify whether the prediction performance exceeded chance levels.”

5. Another major concern that I have relates to the possibility of overfitting. The analysis pipelines are all very complex and the authors have a tiny sample size (47 individuals with autism and 37 controls) which is well known to lead to substantial variation in accuracy (e.g., for a decoding analysis such as the authors pursue here see e.g., Varoquaux NIMG 2018). As far as I can see this could be a possibility for at least two analyses: 1. For the cross-validation of parameters for microscale neural dynamic modelling, the authors use 5-fold cross validation, but not nested cross-validation. The final microcircuit parameters were obtained by averaging across folds. This will almost certainly provide positively biased estimates of predictive performance because a strict training test separation has not been preserved. A related problem is that this hides variation in the parameter estimates due to perturbations due to the cross-validation. I would like to see how variable the estimates are across folds. As noted above, given the sample size I would expect this to be substantial.

We thank the Reviewer for pointing this out. When we performed the five-fold cross-validation with different training and test datasets, mean \pm SD of the estimated microcircuit parameters across the different cross-validation folds were 0.530 ± 0.004 for recurrent excitation/inhibition and 0.325 ± 0.001 for subcortical/external input in controls. Similarly, consistent findings were observed in individuals with autism, with 0.528 ± 0.006 for recurrent excitation/inhibition and 0.325 ± 0.001 for subcortical/external input across folds. To establish robustness of the biophysical network model, we additionally evaluated variation of the relaxed mean-field model across 1000 bootstraps as described above response (#3).

We furthermore highlight potential sample size limitations in the revised *Discussion* (P.12):

“Arguably, the modest sample size of this study poses some limitations on the results from the symptom severity prediction, despite the use of three- and five-fold cross-validations. It, overall, warrants replication of our findings in larger, and ideally transdiagnostic cohorts to also evaluate specificity and to further explore heterogeneity within the autism spectrum itself¹¹⁷⁻¹²⁰.”

6. The symptom prediction (ADOS scores) uses an elastic net. The authors use a fixed regularization parameter setting but again I suspect this could be biased because the authors again do not mention nested cross-validation and again select features that were repeatedly selected across cross-validation folds for the final predictive model.

We thank the Reviewer for the thoughtful comment. The revised paper predicted symptom severity using five-fold nested cross-validation⁵⁴⁻⁵⁷ with regularization parameters ranging from 0.1 (*i.e.*, more to L2-norm) to 1.0 (*i.e.*, L1-norm). Nested cross-validation split the dataset into training (4/5) and test (1/5) partitions, and each training partition is further split into inner training and testing folds using another five-fold cross-validation. The model with lowest overfitting across the inner folds was applied to the test data of the outer fold. After controlling for age, sex, and site from a total of 600 (200 regions \times 3 eigenvectors) features, we selected performant features using an elastic net regularization⁷². ADOS score prediction was performed based on linear regression with these selected features. The procedure was repeated 100 times with different set of training and test data to avoid bias for separating subjects. Indeed, the regularization parameter of 0.6 showed best performance for predicting total ADOS score (mean \pm SD $r = 0.47 \pm 0.06$; mean \pm SD mean absolute error (MAE) = 2.07 ± 0.12 ; permutation-test $p < 0.01$) as well as subscores for social cognition (mean \pm SD $r = 0.43 \pm 0.07$; mean \pm SD MAE = 1.79 ± 0.09 ; permutation-test $p < 0.02$), communication (mean \pm SD $r = 0.57 \pm 0.03$; mean \pm SD MAE = 0.89 ± 0.03 ; permutation-test $p < 0.01$), and marginally for repeated behavior/interest (mean \pm SD $r = 0.33 \pm 0.09$; mean \pm SD MAE = 0.89 ± 0.05 ; permutation-test $p < 0.09$) (**Fig. 4**). Features were selected in premotor, lateral prefrontal and orbitofrontal, lateral and medial temporal, lateral parietal, and posterior cingulate regions. (for results based on other regularization parameters, see **Table S3**).

Fig. 4 | Associations between structural manifolds and autism symptoms. Frequency of the selected brain regions across four ADOS scores reported on the left top. Correlation between actual and predicted ADOS total and subscores are reported. Black line indicates mean correlation and gray lines represent 95% confidence interval for 100 iterations with different training/test dataset. *Abbreviation:* ADOS, Autism Diagnostic Observation Schedule; MAE, mean absolute error.

We updated the *Results* (P.9):

“Specifically, we employed elastic net regularization⁷² with five-fold nested cross-validation (see Methods)^{54–57}. The procedure was repeated 100 times with training and test data compositions to avoid subject selection bias. Using a regularization parameter of 0.6, manifolds spanned by M1–M3 significantly predicted total ADOS score (mean \pm SD $r = 0.47 \pm 0.06$; mean \pm SD mean absolute error (MAE) = 2.07 ± 0.12 ; permutation-test $p < 0.01$) as well as subscores for social cognition (mean \pm SD $r = 0.43 \pm 0.07$; mean \pm SD MAE = 1.79 ± 0.09 ; permutation-test $p < 0.02$), communication (mean \pm SD $r = 0.57 \pm 0.03$; mean \pm SD MAE = 0.89 ± 0.03 ; permutation-test $p < 0.01$), and marginally for repeated behavior/interest (mean \pm SD $r = 0.33 \pm 0.09$; mean \pm SD MAE = 0.89 ± 0.05 ; permutation-test $p < 0.09$) (Fig. 4). Features were selected in premotor, lateral prefrontal and orbitofrontal, lateral and medial temporal, lateral parietal, and posterior cingulate regions. (for results based on other regularization parameters, see Table S3).”

as well as the Methods (P.18):

“We adopted a supervised machine learning framework with nested cross-validation^{54–57} to predict autism symptoms as measured by ADOS⁷¹. We aimed at predicting total ADOS scores, as well as subscores for social cognition, communication, and repeated behavior/interest (Fig. 4 and Table S3). We utilized five-fold nested cross-validation^{54–57} and elastic net regularization⁷² with regularization parameters ranging from 0.1 (i.e., more to L2-norm) to 1.0 (i.e., L1-norm). Nested cross-validation split the dataset into training (4/5) and test (1/5) partitions, and each training partition was further split into inner training and testing folds using another five-fold cross-validation. The model with lowest overfitting across the inner folds was applied to the test partition of the outer fold. After controlling for age, sex, and site from a total of 600 (200 regions \times 3 manifolds) features, we selected performant features using elastic net regularization. ADOS score prediction leveraged linear regression with the selected features. The procedure was repeated 100 times with different training and test partitions.”

Finally, we performed an additional sensitivity analysis that repeated the predictive approach using three-fold cross-validation. While performance was marginally lower, it was still promising. These findings are now presented in the Results and Methods as follows (P. 10, 18):

“Performing the same analyses using three-fold nested cross-validation, we found largely consistent prediction results. Indeed, the regularization parameter of 0.7 and 0.8 showed good performance for predicting ADOS scores, and features were primarily selected in lateral and medial prefrontal, lateral parietal, and lateral temporal regions (Table S4).”

“We repeated the prediction analysis using three-fold nested cross-validation, so that each fold includes more training and test data (Table S4).”

7. From a conceptual level how do the authors reconcile their manifold learning approach with a parcellation approach that follows? The results are not necessarily complementary with one another since the manifold learning approach estimates multiple *spatially overlapping* patterns of connectivity. The parcellation approach that follows necessarily involves averaging these signals into piece-wise constant parcels which potentially mixes signals from the large-scale cortical gradients. See Haak et al. NIMG 2018, 2020, for further discussion on this issue.

We are happy to better motivate our approach and to differentiate it from parcellation approaches, and are thankful for the suggested references. Please see the revised Introduction (P. 3):

“Macroscale brain organization is increasingly studied using manifold learning techniques that project high dimensional connectomes into low dimensional representations. These approaches allow representing continuous changes in connectivity across the cortical surface, and can incorporate multiple, potentially overlapping, connectivity gradients^{36,37}. As such, they provide an alternative perspective to widely used parcellation approaches that place discrete boundaries and that average connectivity measures within each parcel, which may potentially mix the signals from different large-scale gradients.”

8. Other minor points: I am not convinced the authors have done enough to address site effects. For example, I would also like to see symptom prediction models estimated separately for each site (after addressing the concerns about overfitting above)

Following the Reviewer’s comment, we performed symptom severity prediction for each site using the suggested approach. Although each site contains small number of subjects ($n = 20$ for NYU, $n = 18$ for TCD), we found consistent prediction results with similar optimal regularization parameters (0.7 for NYU, 0.4 for TCD). We reported the results in Table S5 and S6.

We updated the *Results* (P.10):

“We repeated symptom severity prediction for each site separately. Although each site contains small number of subjects ($n = 20$ for New York University Langone Medical Center (NYU), $n = 18$ for Trinity College Dublin (TCD)), we found consistent prediction results with similar optimal regularization parameters (Table S5 and S6).”

as well as the *Methods* (P.18):

“a) Site effects. We also repeated the symptom severity prediction for each site using five-fold nested cross-validation to assess whether the prediction results are consistent across different sites (Table S5 and S6).”

9. The authors do not do any statistical tests to determine whether the dynamical model for connectivity predicted functional connectivity better than the baseline model. Given the small sample size, I doubt whether a difference of $r = 0.26$ on the test set is really different from 0.23 (see p5)

At a high resolution, the biophysical model achieved a correlation between empirical and simulated functional connectivity of $r = 0.50$ for the training and of 0.26 for the test data, which was higher than the baseline correlation between structural connectivity and empirical functional connectivity ($r = 0.25$ for training, $r = 0.23$ for test). Due to the computational complexity (see *above*), we could not assess statistical significance for this improvement in predicting of functional connectivity with the biophysical model when the 200×200 data was used. Instead, we applied the same dimensionality reduction procedure and bootstraps as described above. Here, we calculated correlation coefficients between structural connectivity and empirical functional connectivity 1,000 times to construct a baseline null distribution. If the mean correlation between empirical and simulated functional connectivity across bootstraps did not belong to 95% of the baseline null distribution, we considered the model significantly improved the correspondence between empirical and simulated functional connectivity compared to a baseline. Only one side of the null distribution was considered. For controls, the optimal model indeed predicted functional connectivity ($r = 0.68$ for training and $r = 0.65$ for test), outperforming baseline correlations between structural and functional connectivity ($r = 0.53$, $p < 0.001$ for training and $r = 0.49$, $p < 0.001$ for test) (**Fig. S8B**). Significant improvements were also observed in individuals with autism (**Fig. S8B**; $p < 0.001$ for both training and test), indicating that the biophysical model significantly improved the association between structural and functional connectivity.

B. Functional community-wise computational simulations

Fig. S8 | Microcircuit parameters and biophysical simulations. (B) Linear correlations between empirical functional connectivity (FC) and structural connectivity (SC), and empirical and simulated FC for controls and individuals with autism based on functional communities⁶². Black lines indicate mean correlation and gray lines represent 95% confidence interval across 1,000 bootstrapping.

We updated the *Results* (P.6):

“We assessed statistical significance for the improvement in predicting of functional connectivity with the biophysical model compared to a baseline model based on structural connectivity by conducting bootstraps using dimensionality reduced structural and functional connectomes based on functional communities⁶² (see *Methods*). The correlation coefficients between

empirical and simulated functional connectivity outperformed the corresponding baseline correlations between structural and functional connectivity in both controls and individuals with autism (Fig. S8B; $p < 0.001$ for both training and test)."

as well as the *Methods* (P.16-17):

"We also calculated correlation coefficients between structural connectivity and empirical functional connectivity 1,000 times to construct a null distribution. If the mean correlation between empirical and simulated functional connectivity across bootstraps did not belong to 95% of the baseline null distribution, we considered the model significantly improved the correspondence between empirical and simulated functional connectivity compared to baseline. Only one side of the null distribution was considered."

REVIEWER 4:

The paper by Park et al. carries out connectome study comparing autism cases and neurotypical controls and then relates the connectome changes to gene expression data from the Allen brain atlas. The transcriptome analysis is potentially interesting, but it needs additional supporting evidence and validation in an independent gene expression dataset.

We thank the Reviewer for their interest and for the constructive comments.

1. Major points: 1. It is not clear how the gene decoding analysis was done and therefore whether the approach is appropriate. Was the t-statistics map correlated with gene expression data from each individual from the Allen Brain atlas, or was gene expression averaged across individuals? If averaged, how is inter-individual variation taken into account statistically?

To clarify our approach, we used Neurovault which implements mixed-effect analysis to estimate associations between the input t-statistic map and the genes of the Allen Institute for Brain Sciences (AIBS) donor brains, yielding the gene symbols associated with the input t-statistic map. For each gene, a linear model fits the input map to each of the donor brains. A one sample t-test is applied to assess how the relation between the gene expression and input t-statistic map are consistent across the donated brains. We updated the *Results* (P.7):

"We performed transcriptomic association analysis and developmental and disease enrichment analysis to explore neurobiological underpinnings of the above macroscale manifold findings (Fig. 3A). Specifically, we correlated the multivariate change pattern with post-mortem gene expression data of six donors from the Allen Institute for Brain Sciences (AIBS) ^{65,66}."

as well as the *Methods* (P.17):

"For each gene, a linear model fits the input map to each of the six brains donated to the AIBS. A one sample t-test assessed whether the relation between the gene expression and input t-statistic map are consistent across the donated brains."

To incorporate inter-individual variation as suggested by the Reviewer, we included an additional step and only selected those genes that were consistently expressed across donors (*i.e.*, genes with an average inter-donor correlation of gene expression pattern of $r > 0.5$) for the developmental enrichment analysis (see *Methods*) ⁶⁷. This analysis confirmed associations between the multivariate pattern of autism-related structural manifold distortions and genes expressed in early childhood and adolescence, as well as early infancy and young adulthood, in thalamic and cortical areas (**Fig. 3B**). While these genes were also expressed in the cerebellum in early developmental and amygdala in later developmental stages, they were not significantly expressed in as the striatum or in the hippocampus.

Fig. 3 | Transcriptomic analysis to identify gene expression patterns. (A) Schema to relate multivariate manifold distortions with gene expression patterns and to perform a cell-type specific gene expression analysis. (B) Developmental enrichment, showing strong associations with cortex and thalamus during early childhood and adolescence, as well as early infancy and young adulthood. The size of hexagon rings represents the proportion of genes specifically expressed in particular tissue at particular developmental stage. Varying stringency for enrichment with respect to specificity index threshold (pSI) are represented by the size of hexagons going from least specific (outer hexagons) to most specific (center hexagons) (pSI = 0.05, 0.01, 0.001, and 0.0001, respectively)⁵³. Colors represent the false discovery rate (FDR)-corrected p-values. The bar plot on the left represents the log transformed p-values, averaged across all brain structures.

We updated the *Results* (P.7-8):

“Among the significantly associated genes, we selected only those that were consistently expressed across donors ($r > 0.5$)⁶⁷ (Data S1). We fed those into a developmental gene expression analysis, which highlights developmental time windows across brain regions in which these genes are expressed (see Methods)⁵³. This analysis highlighted associations between the multivariate pattern of autism-related structural manifold distortions and genes expressed in early childhood and adolescence, as well as early infancy and young adulthood, in thalamic and cortical areas (Fig. 3B). While these genes were also expressed in the cerebellum in early developmental and amygdala in later developmental stages, they were not significantly expressed in the striatum or hippocampus.”

And the *Methods* (P.17)

“We further examined which of the significant genes were consistently expressed across donors using *abagen* (<https://github.com/rmarkello/abagen>)⁶⁷. For each gene, we correlated the whole-brain gene expression map between all pairs of donors, and considered only genes with an average inter-donor $r > 0.5$ for subsequent analyses.”

2. Are the observed correlations replicable in an independent dataset (e.g., GTEx)?

We thank the Reviewer for this suggestion. As proposed, we additionally validated our transcriptomic association results using the Genotype-Tissue Expression (GTEx) database (<https://www.gtexportal.org/home/>). We used the multi-gene query function, which calculates transcripts per million (TPM) of each gene to quantify the degree of enrichment to a given brain structure. We entered the top 30 ranked genes derived from Neurovault to the multi-gene query of GTEx, and found that tissue types of frontal cortex (BA9) and cortex showed highest TPM (25.65 and 20.77, respectively) compared to other brain structures (Fig. S10A). As GTEx does not provide the same brain structures with cell-type specific expression analysis (CSEA), we could not quantify how the genes expressed in thalamus. On the other hand, we could find strong gene expressions in cortical areas, largely consistent with our transcriptomic association results in Fig. 3B.

Fig. S10 | Gene enrichment using Genotype-Tissue Expression (GTEX) and cell-type specific expression analysis (CSEA). (A) Top 30 ranked genes derived from Neurovault were fed into the multi-gene query of GTEX and transcripts per million (TPM) of each gene was calculated for different brain structures. Each row represents genes and column represents brain structures. The average TPM value across genes for each brain structure is reported on the top. *Abbreviations:* BA, Brodmann area.

We updated the *Results* (P.8):

“We furthermore validated the transcriptomic association results using the Genotype-Tissue Expression (GTEX) database (<https://www.gtexportal.org/home/>), and we confirmed that the genes highly associated with multivariate manifold changes were strongly expressed in cortical areas (Fig. S10A).”

as well as the *Methods* (P.17-18):

“We also replicated the gene enrichment results with a different database, we additionally performed transcriptomic association analysis using the Genotype-Tissue Expression (GTEX) database (<https://www.gtexportal.org/home/>; Fig. S10A). We used the multi-gene query function, which calculates transcripts per million (TPM) of each gene to quantify the degree of enrichment to a given brain structure. We entered the top 30 ranked genes derived from Neurovault to the multi-gene query of GTEX.”

3. The authors need to address the possibility that the observed correlations may be driven by cell type composition differences between brain regions. Are the significantly correlated genes enriched for cell-type specific genes?

We thank the Reviewer for this thoughtful comment. To address cell-type specific gene enrichment, we compared the genes associated to multivariate connectome manifold changes with cell-type specific expression analysis (CSEA) proposed in prior work^{69,70}. The distinct cell types include excitatory and inhibitory neuronal subtypes in the cortex, and non-neuronal cells of endothelial cells, smooth muscle cells or pericytes, astrocytes, oligodendrocytes and their precursor cells, and microglia^{69,70}. For each cell type, we calculated the overlap ratio how many genes expressed for manifold changes are included in each cell-type specific genes. The highest overlap was observed in the excitatory neurons (mean \pm SD across 13 cell subtypes = $2.56 \pm 0.65\%$) and astrocytes (2.11%) relative to others (inhibitory neurons: mean \pm SD across 11 cell subtypes = $0.91 \pm 0.48\%$; endothelial cells: 1.96%; pericytes, oligodendrocytes and their precursor cells, microglia: 0%). These findings are now provided in the revised *Results* and *Methods* (P.9, 18):

“We further compared the genes associated to multivariate connectome manifold changes with distinct cell types proposed in prior work^{69,70}. For each cell type, we calculated the overlap ratio how many genes expressed for manifold changes are included in each cell-type specific genes (Fig. 10B). The highest overlap was observed in the excitatory neurons (mean \pm SD

across 13 cell subtypes = $2.56 \pm 0.65\%$) and astrocytes (2.11%) relative to others (inhibitory neurons: mean \pm SD across 11 cell subtypes = $0.91 \pm 0.48\%$; endothelial cells: 1.96%; pericytes, oligodendrocytes and their precursor cells, microglia: 0%).”

“To address cell-type specific gene enrichment, we compared the genes associated to multivariate connectome manifold changes with CSEA proposed in prior work^{69,70}. The distinct cell types include excitatory and inhibitory neuronal subtypes in the cortex, and non-neuronal cells of endothelial cells, smooth muscle cells or pericytes, astrocytes, oligodendrocytes and their precursor cells, and microglia^{69,70}. For each cell type, we calculated the overlap ratio how many genes expressed for manifold changes are included in each cell-type specific genes (Fig. 10B).”

B. Cell-type specific expression

Fig. S10 | Gene enrichment using Genotype-Tissue Expression (GTEx) and cell-type specific expression analysis (CSEA). (B) The overlap ratio between the genes expressed for manifold changes and each cell-type specific genes. Error bars represent standard error of the mean for sub-cell-types. Only excitatory and inhibitory neurons have sub-cell-types.

Reviewer #1 (Remarks to the Author):

The authors have comprehensively and thoroughly addressed my earlier suggestions. The new analyses have strengthened the revised version and leave no stone unturned. This is a timely, novel and important paper. While edge-level effects may be washed out and diluted by the extensive dimensionality reduction, potentially explaining the small effects, this sounds like a very reasonable way forward to overcome the computational limitations of the model.

Minor consideration: It may be helpful to briefly comment on the practical implications of the magnitude of between-group differences in the microcircuit parameters. The differences appear to be quite subtle in Fig. 8C, particularly when contrasted against the magnitude of the between group differences in structural connectivity (Fig S6). Even showing a boxplot for each functional community, with individual data points indicated, would be helpful to understand variation across individuals.

Reviewer #2 (Remarks to the Author):

All my comments have been addressed. The manuscript has much improved following this round of revisions.

Reviewer #3 (Remarks to the Author):

The authors have addressed many of the concerns I raised in my original review, particularly relating to the clarity of the manuscript and the inclusion of nested cross-validation for the elastic net prediction is a welcome addition.

That said, I still think that the very small sample is a major limitation to the impact of this work. Whilst I defer the final decision on this issue to the editor, I would like to point out that this does have important implications on the analyses presented, for example, as I expected the parameter estimates across cross-validation folds are quite unstable - correlations of 0.3-0.5 across folds which is substantial considering that most of the training data is shared across folds. I do not consider that this is satisfactorily addressed and acknowledged in the text and is currently only given a single sentence in the discussion.

I am also not convinced by the significance testing framework that the authors employ to test whether their approach outperforms the baseline approach. While I am sympathetic to difficulties in computational scaling for permutation testing, the fact remains that the authors are not testing the correct null to substantiate the claims in their manuscript (e.g. they still state: "the optimal model indeed predicted functional connectivity (linear correlation coefficient $r \sim 0.5$ for training and $r \sim 0.26$ for test data), outperforming baseline correlations between structural and functional connectivity ($r \sim 0.25$ for training and $r \sim 0.23$ for test data)").

Reviewer #4 (Remarks to the Author):

The authors have satisfactorily addressed my questions. One minor issue that still needs to be addressed: the cellular composition analysis needs a statistical test to show whether the overlap with cell-type specific genes is more than expected by chance (eg. Fisher test, hypergeometric test).

REVIEWER 1:

The authors have comprehensively and thoroughly addressed my earlier suggestions. The new analyses have strengthened the revised version and leave no stone unturned. This is a timely, novel and important paper. While edge-level effects may be washed out and diluted by the extensive dimensionality reduction, potentially explaining the small effects, this sounds like a very reasonable way forward to overcome the computational limitations of the model.

We thank the Reviewer for the positive appraisal of our response, and for appreciating the significance and timeliness of the work.

1. Minor consideration: It may be helpful to briefly comment on the practical implications of the magnitude of between-group differences in the microcircuit parameters. The differences appear to be quite subtle in Fig. S8C, particularly when contrasted against the magnitude of the between group differences in structural connectivity (Fig S6). Even showing a boxplot for each functional community, with individual data points indicated, would be helpful to understand variation across individuals.

We thank the Reviewer for these suggestions. We reported the microcircuit parameters using box plots with individual data points across 1,000 bootstraps according to functional communities in **Fig. S8C**, as well as radar plots.

Fig. S8 | Microcircuit parameters and biophysical simulations. (C) (Left) Microcircuit parameters of controls and individuals with autism. (Middle) The functional community-wise⁶³ stratifications of the parameters are presented with radar plots. Black lines indicate controls normalized to mean zero and standard deviation of one, and red lines indicate individuals with autism normalized according to controls. Solid and dash lines represent mean and standard deviation of the parameters across 1,000 bootstraps, respectively. (Right) The scatter plots represent microcircuit parameters across 1,000 bootstraps. Each dot is the estimated parameter for each bootstrap. The white horizontal lines indicate mean of the parameter across bootstraps and gray vertical lines are standard deviation. Brain networks with significant between-group differences are represented with asterisks.

Although differences in microcircuit parameters between individuals with autism and controls indeed appeared small compared to those when comparing edge-wise structural connectivity, they were significant after multiple comparisons correction (Fig. S8C) and overall correlated with connectome manifold findings (Fig. 2B). However, please note that the biophysical network modeling approach harnesses structural and functional connectivity synergistically, not just one of these components. As such, structural connectivity comparisons and microcircuit models may have differential sensitivity. Indeed, while the former approach is optimized to detect differences in structural connectivity, the latter capitalizes on structure-function coupling and thus provides a useful bridge between perturbed

macroscale connectivity on the one hand, and microcircuit dysfunction on the other hand. These points are now further clarified on *P.11-12*.

“Mapping connectivity alterations at the edge-level revealed widespread alterations in structural connectivity between individuals with autism and controls, supporting the sensitivity of these analyses to map autism-related network alterations. On the other hand, the microcircuit modeling approach harnessed both structural and functional connectivity information synergistically, making a direct comparison of the relative sensitivity of edge-level comparison vis a vis the modeling approach difficult. While the former is optimized to detect differences in structural connectivity between groups, the latter reveals additional insights into structure-function coupling and also provides a useful bridge between perturbed macroscale connectivity, on the one hand, and microcircuit dysfunction, on the other.”

We further added meaning of the differences in microcircuit parameters between individuals with autism and controls in the *Discussion (P.11)*:

“Importantly, comparing microcircuit maps between individuals with autism and neurotypical controls suggested a relatively diffuse pattern of local microcircuit parameter changes, rather than a pattern of cortical microcircuit alterations along a specific direction. In healthy individuals, inter-regional variations in recurrent excitation/inhibition and subcortical/external input follow sensory-fugal hierarchical gradients, previously established with resting-state functional connectivity mapping and analysis of myelin-sensitive MRI contrasts¹⁸. Our computational modeling findings, showing atypical microcircuit parameters in somatosensory but also higher-order default mode networks in individuals with autism relative to controls, suggest a broad microcircuit imbalance in autism that is not limited to a specific network. Our results also indicate that these microcircuit imbalances impact multiple stages of the cortical hierarchy, a pattern consistent with prior resting-state fMRI analysis, and with the broad phenotypical correlates of autism that encompass both sensory deficits as well as atypical features of higher-order cognition^{6,8,16,114}. In our work, directly correlating connectome-wide manifold distortions with microcircuit parameters indicated associations with increased recurrent excitation/inhibition and excitatory subcortical drive, a finding that was especially marked in idiosyncratic areas with strong laminar differentiation. Findings were robust after controlling for spatial autocorrelations using non-parametric spin tests, collectively suggesting the links between macro and microscale neuronal alterations in autism.”

REVIEWER 2:

All my comments have been addressed. The manuscript has much improved following this round of revisions.

We thank the Reviewer for appreciating our revisions and manuscript.

REVIEWER 3:

The authors have addressed many of the concerns I raised in my original review, particularly relating to the clarity of the manuscript and the inclusion of nested cross-validation for the elastic net prediction is a welcome addition.

We thank the Reviewer for appreciating our efforts to address their comments, and we are grateful for the additional suggestions to strengthen our approach

1. That said, I still think that the very small sample is a major limitation to the impact of this work. Whilst I defer the final decision on this issue to the editor, I would like to point out that this does have important implications on the analyses presented, for example, as I expected the parameter estimates across cross-validation folds are quite unstable - correlations of 0.3-0.5 across folds which is substantial considering that most of the training data is shared across folds. I do not consider that this is satisfactorily addressed and acknowledged in the text and is currently only given a single sentence in the discussion.

To clarify, the reported values of $0.530 \pm 0.004 / 0.528 \pm 0.006$ and $0.325 \pm 0.001 / 0.325 \pm 0.001$ for controls/autism do not refer to the cross-correlations of the model parameters across the cross-validation folds. Instead, these represent the mean \pm SD of the absolute values of microcircuit parameters (*i.e.*, recurrent excitation/inhibition and subcortical/external input) across the cross-validation folds.

When computing correlations of these parameters across five folds, we obtained high mean \pm SD correlation coefficients of $r = 0.79 \pm 0.04$ for recurrent excitation/inhibition and $r = 0.91 \pm 0.03$ for subcortical/external input for controls, and 0.74 ± 0.05 and 0.89 ± 0.03 for individuals with autism. These results support stability of the model derived parameters.

We have thus clarified and expanded the *Results* (P.6):

“Estimated microcircuit parameters (Fig. 2B) were relatively stable across cross-validations, with a mean \pm SD of the parameter values across the five folds of 0.530 ± 0.004 for recurrent excitation/inhibition and 0.325 ± 0.001 for subcortical/external input in controls, and 0.528 ± 0.006 for recurrent excitation/inhibition and 0.325 ± 0.001 for subcortical/external input in autism. The product-moment correlations of the microcircuit parameters across the cross-validation folds were mean \pm SD of 0.79 ± 0.04 for recurrent excitation/inhibition and 0.91 ± 0.03 for subcortical/external input for controls, and 0.74 ± 0.05 and 0.89 ± 0.03 for individuals with autism, indicating the robustness of the biophysical model.”

as well as the *Methods* (P.17):

“The robustness of the estimated model parameters was assessed by calculating cross-correlations of recurrent excitation/inhibition and subcortical/external input across five folds.”

We are nevertheless appreciative of the Reviewer’s suggestion to expand on our *Discussion* on the need to increase sample sizes in future work (P.13):

“Despite assessing cross-site variability, implementing bootstrap-based assessment, and running cross-validations where appropriate, the modest sample size potentially limits the generalization of our results. Of note, although we leveraged ABIDE, the currently largest neuroimaging database in autism that is openly accessible, the restriction to individuals who had dMRI, rs-fMRI, and structural MRI data of adequate quality reduced the available sample size. Prior conceptual and simulation findings suggested that a small sample size yields high variance in predictive errors and emphasized the value of confirming findings in independent and large-scale datasets where possible, not to underestimate prediction error¹²⁵. This can benefit from the aggregation and sharing of more open datasets, ultimately strengthening the generalizability of diagnostic biomarkers^{125–127}. A further avenue may also involve the study of transdiagnostic cohorts, which would not only provide additional consolidation of our findings, but also help to evaluate how specific this pattern is to autism, and to further explore inter-individual heterogeneity within the condition^{128–131}.”

2. I am also not convinced by the significance testing framework that the authors employ to test whether their approach outperforms the baseline approach. While I am sympathetic to difficulties in computational scaling for permutation testing, the fact remains that the authors are not testing the correct null to substantiate the claims in their manuscript (e.g. they still state: “the optimal model indeed predicted functional connectivity (linear correlation coefficient $r \sim 0.5$ for training and $r \sim 0.26$ for test data), outperforming baseline correlations between structural and functional connectivity ($r \sim 0.25$ for training and $r \sim 0.23$ for test data)”.

We thank the Reviewer for the comment. In our previous manuscript, we evaluated the relaxed mean-field model with 1,000 bootstraps to test whether simulated functional connectivity (FC’) has a stronger correlation with empirical functional connectivity (FC) compared to the correlation between structural connectivity (SC) and FC. This evaluation indicated that $\text{corr}(FC', FC)$ was higher than $\text{corr}(SC, FC)$ in 1000/1000 bootstraps (*i.e.*, 100%).

In response to the Reviewer's comment, we implemented and reported two additional evaluations. First, we reported the number of cross-validation folds in which $\text{corr}(FC', FC)$ was higher than $\text{corr}(SC, FC)$. Here, we observed that the model derived measures were higher than baseline measures in 5/5 folds, for both training and test data.

Next, we performed 1,000 permutation tests by randomly assigning elements of FC' and SC . We calculated the differences in correlations (i.e., $\text{corr}(pFC', FC) - \text{corr}(pSC, FC)$, where p denotes permutation) 1,000 times and constructed a null distribution. If the real difference in correlations (i.e., $\text{corr}(FC', FC) - \text{corr}(SC, FC)$) did not belong to 95% of the null distribution, we considered $\text{corr}(FC', FC)$ is significantly higher than $\text{corr}(SC, FC)$. Only one side of the null distribution was considered. We found that the optimal model predicted functional connectivity significantly higher than the baseline ($p < 0.001$ for both training and test data).

We updated the *Results (P.6)*:

“The optimal model predicted functional connectivity (product-moment correlation coefficient $r \sim 0.5$ for training and $r \sim 0.26$ for test data) nominally higher than the corresponding baseline correlations between structural and functional connectivity ($r \sim 0.25$ for training and $r \sim 0.23$ for test data) (Fig. 2A and S8A). We assessed improvements in predicting functional connectivity with the biophysical model compared to a baseline model based on structural connectivity with three different approaches. First, we observed that the model-driven correlations (i.e., between empirical and simulated functional connectivity) were higher than baseline correlations (i.e., between structural and empirical functional connectivity) in all 5/5 folds for both training and test data. Second, we performed 1,000 permutation tests by assigning elements of structural and simulated functional connectivity (see Methods). We found that the empirical and simulated functional connectivity showed significantly higher correlation compared to structural and empirical functional connectivity ($p < 0.001$ for both training and test data). Finally, we conducted 1,000 bootstraps using dimensionality reduced structural and functional connectomes based on functional communities⁶³ (see Methods). The correlation coefficients between empirical and simulated functional connectivity exceeded corresponding baseline correlations between structural and functional connectivity (Fig. S8B; $p < 0.001$ for both training and test). Together, these findings suggest that the biophysical model provided improvements in predicting functional connectivity relative to using baseline structural connectivity.”

as well as the *Methods (P.17)*:

“To assess whether the biophysical model predicted FC better than baseline SC, we implemented three different approaches. First, we counted the number of cross-validation folds, where the correlation between FC and FC' was higher than the correlation between SC and FC. Second, we performed 1,000 permutation tests by randomly assigning elements of FC' and SC . We calculated the differences in correlations (i.e., $\text{corr}(pFC', FC) - \text{corr}(pSC, FC)$, where p denotes permutation) 1,000 times and constructed a null distribution. If the real difference in correlations (i.e., $\text{corr}(FC', FC) - \text{corr}(SC, FC)$) fell outside of the 95% confidence interval of the null distribution, we considered the correlation between empirical and simulated functional connectivity to be significantly higher than the correlation between structural and empirical functional connectivity. Only one side of the null distribution was considered. Finally, we evaluated variation of the relaxed mean-field model using 1,000 bootstraps by randomly sampling 90% of subjects with replacement within each group. To reduce the computational complexity, we first reduced the dimensionality of structural and functional connectomes based on seven established functional communities⁶³. Then, we repeated estimating microcircuit parameters 1,000 times with the dimensionality reduced connectomes (Fig. S8B). We calculated correlation coefficients between SC and FC 1,000 times and constructed a distribution of the baseline correlations. If the mean correlation between FC and FC' across bootstraps falls outside of the 95% of the baseline distribution, then we considered the model to significantly improve the correspondence between empirical and simulated functional connectivity compared to baseline. Only one side of the null distribution was considered.”

REVIEWER 4:

The authors have satisfactorily addressed my questions.

We thank the Reviewer for appreciating our efforts to address the Reviewer’s concerns, and are happy to include further suggestion.

One minor issue that still needs to be addressed: the cellular composition analysis needs a statistical test to show whether the overlap with cell-type specific genes is more than expected by chance (e.g., Fisher test, hypergeometric test).

As suggested, we implemented permutation tests to verify whether the overlap ratios between genes associated to connectome manifold changes and those of distinct cell types are significant. Among all cell-type specific genes, we assigned the genes to each cell type with the same gene length. Then, we calculated the overlap ratio between the genes expressed for manifold changes and the permuted cell-type specific genes. For each cell type, we calculated the overlap ratio 1,000 times and constructed a null distribution. If the real overlap ratio did not belong to 95% of the null distribution, it was deemed significant. Multiple comparisons across different cell types were corrected using FDR, showing a marginal effect for excitatory neurons (FDR = 0.1).

We updated the *Results* (P.9):

“Cell-type specific expression analysis indicates several cell types showing a similar expression profile with positive overlap ratio. Highest and marginally significant overlap was observed for the excitatory neurons (mean \pm SD across 13 cell subtypes = $22.36 \pm 4.16\%$; FDR = 0.1; see Methods) relative to others (astrocytes: 11.11%; inhibitory neurons: mean \pm SD across 11 cell subtypes = $7.47 \pm 4.21\%$; endothelial cells: 14.29%; pericytes, oligodendrocytes and their precursor cells, microglia: 0%).”

as well as the *Methods* (P.19):

“To assess the significance of the overlap ratio, we performed 1,000 permutation tests. Among all cell-type specific genes, we assigned the genes to each cell type with the same gene length. Then, we calculated the overlap ratio between the genes expressed for manifold changes and the permuted cell-type specific genes. For each cell type, we calculated the overlap ratio 1,000 times and constructed a null distribution. If the real overlap ratio did not belong to 95% of the null distribution, it was deemed significant. Multiple comparisons across different cell types were corrected using the FDR procedure¹⁴⁹.”

B. Cell-type specific expression

Fig. S10 | Gene enrichment using Genotype-Tissue Expression (GTEx) and cell-type specific expression analysis (CSEA). (B) The overlap ratio between the genes expressed for manifold changes and each cell-type specific genes. Error bars represent standard error of the mean for sub-cell-types. Only excitatory and inhibitory neurons have sub-cell-types.

Reviewer #1 (Remarks to the Author):

No further comments. Thanks for addressing the remaining comment and well done!

Reviewer #3 (Remarks to the Author):

I think the authors have made best efforts to address my comments and the more open acknowledgement of the sample size is a particularly welcome addition. I have no further comments.

Reviewer #4 (Remarks to the Author):

I am happy with the author's response, they have addressed my suggestion from the last round of review and I have no further comments.